# Myosin V executes steps of variable length via structurally constrained diffusion

David Hathcock[1]*, Riina Tehver[2], Michael Hinczewski[3]*, D Thirumalai[4]

[1]Department of Physics, Cornell University, Ithaca, United States; [2]Department of Physics and Astronomy, Denison University, Granville, United States; [3]Department of Physics, Case Western Reserve University, Cleveland, United States; [4]Department of Chemistry, University of Texas, Austin, United States

**Abstract** The molecular motor myosin V transports cargo by stepping on actin filaments, executing a random diffusive search for actin binding sites at each step. A recent experiment suggests that the joint between the myosin lever arms may not rotate freely, as assumed in earlier studies, but instead has a preferred angle giving rise to structurally constrained diffusion. We address this controversy through comprehensive analytical and numerical modeling of myosin V diffusion and stepping. When the joint is constrained, our model reproduces the experimentally observed diffusion, allowing us to estimate bounds on the constraint energy. We also test the consistency between the constrained diffusion model and previous measurements of step size distributions and the load dependence of various observable quantities. The theory lets us address the biological significance of the constrained joint and provides testable predictions of new myosin behaviors, including the stomp distribution and the run length under off-axis force.

*For correspondence:
dch242@cornell.edu (DH);
mxh605@case.edu (MH)

Competing interests: The authors declare that no competing interests exist.

## Introduction

Molecular motors are cellular machines that function by converting chemical energy into mechanical work (*Schliwa and Woehlke, 2003*). Motors play key roles in many intracellular biological processes, including signaling and the transport of cargo (*Sun and Goldman, 2011*). Members of the myosin superfamily, one class of molecular motors, perform these functions by binding to actin filaments and generating energy through ATP hydrolysis. Myosin V, a dimeric transport motor, is composed of two stiff polymer chains joined at a pivot with an actin-binding head at the end of each chain (*Reck-Peterson et al., 2000*). The motor walks forward along the actin, stepping hand-over-hand, by alternating head detachment, with the free head performing a diffusive search for actin binding sites during each step (*Shiroguchi and Kinosita, 2007*). Such unidirectional motility requires coordination between the two heads with preferential detachment of the rear head, the so called 'gating' mechanism, which is regulated by the strain within the lever arms while the heads are bound to actin (*Veigel et al., 2002*; *Veigel et al., 2005*; *Purcell et al., 2005*; *Sakamoto et al., 2008*). Myosin V propels itself toward the plus (barbed) end of the actin using two changes in the lever arm orientation. The power stroke, executed by an actin-bound head, swings the lever arm forward, while the recovery stroke, executed during diffusion, returns the lever to its original orientation, which favors binding to forward actin sites (*Shiroguchi et al., 2011*). Most frequently, myosin V takes $\approx 74$ nm steps, roughly equal in length to the actin helical pitch, but shorter and longer steps also occur (*Yildiz et al., 2003*). The near correspondence of the step size with the pitch, as well as the narrow step distribution, allows myosin V to approximately maintain its azimuthal orientation with respect to the actin over multiple steps (*Sun and Goldman, 2011*). Mutant myosins with altered lever arms show a linear relation between mean step size and arm length (*Sakamoto et al., 2005*; *Oke et al.,*

*2010*). Under moderate backward force myosin V remains highly processive, up to a stall force ≈ 1.9 - 3 pN, (*Mehta et al., 1999*; *Veigel et al., 2002*; *Kad et al., 2008*; *Uemura et al., 2004*; *Cappello et al., 2007*; *Gebhardt et al., 2006*), at which the mean velocity of the motor goes to zero.

While myosin V is one of the most extensively studied motors, new functional features continue to be discovered as the spatiotemporal resolution of experimental imaging improves. Most recently *Andrecka et al. (2015)* achieved simultaneous millisecond temporal and nanometer spatial resolution with interferometric scattering microscopy in which the position of the gold nanoparticle attached to one of the motor heads was used to track the diffusion of the free head during its step. This measurement and recent electron micrographs of freely floating myosin taken by *Takagi et al. (2014)* indicate that the joint between the myosin V lever arms does not rotate freely, but instead has structural constraints giving rise to a preferred inter-arm joint angle. The presence of a joint constraint seems to contradict previous diffusion measurements by *Dunn and Spudich (2007)* and other experiments (*Shiroguchi and Kinosita, 2007*; *Fujita et al., 2012*; *Beausang et al., 2013*) which indicated a freely rotating joint. Further, a number of theoretical and numerical studies based on free diffusion models have been remarkably successful in quantitatively describing various aspects of myosin V motility (*Hinczewski et al., 2013*; *Craig and Linke, 2009*; *Mukherjee et al., 2017*).

To address these apparent conflicts, we extend a minimal model of myosin V previously introduced by *Hinczewski et al. (2013)*, incorporating a constraint on the relative orientation of the two lever arms. The model combines a coarse-grained polymeric description of the diffusive search (*Thirumalai and Ha, 1998*) with the reaction network of discrete states taken by the motor heads during the mechanochemical stepping cycle (*Vilfan, 2005a*; *Bierbaum and Lipowsky, 2011*; *Sumi, 2017*). The large persistence length of the myosin V lever arms allows us to derive an approximate but accurate semi-analytical expression for the equilibrium distribution of positions occupied by the free head during the diffusive search. The kinetic network accounts for not only forward steps, but also foot stomps (the head reattaching near the site of detachment) and backward steps which have been observed experimentally (*Kodera et al., 2010*) and become more prominent as the resistive force increases. In contrast to the simplified model of *Hinczewski et al. (2013)*, here we include the full set of available binding sites on the double-helical actin filaments, enabling a description of the distributions of steps and stomps taken by myosin V. The backward force applied by the load induces conformational changes in the lever arms, altering the diffusive search for binding sites and the associated binding probabilities. The effects of the magnitude of the resistive force and direction are easily incorporated into the theory. We supplement the analytical theory with Brownian dynamics (BD) simulations of myosin stepping dynamics, the results of which largely concur with analytical predictions. In addition to addressing the constrained diffusion hypothesis, our model provides the most comprehensive accounting to date of the full range of sub-stall experimental data, including step-size distributions and the load dependence of several physical observables.

The polymer model gives direct insights into the connection between structural features of myosin V, including the inter-arm joint constraint, and both the diffusive search and kinetics of the motor. With a joint constraint, our model predicts diffusion profiles similar to that observed by *Andrecka et al. (2015)*. By computing the changes in diffusion as the constraint strength is varied, we estimate upper and lower bounds on the constraint energy. Fitting the model to experimental measurements of myosin V step distributions (*Yildiz et al., 2003*; *Sakamoto et al., 2005*; *Oke et al., 2010*) and the force dependence of the backward-to-forward step ratio (*Kad et al., 2008*) and mean run length/velocity (*Mehta et al., 1999*; *Uemura et al., 2004*; *Clemen et al., 2005*; *Gebhardt et al., 2006*; *Kad et al., 2008*), we confirm the consistency of the constrained diffusion picture with previous experimental data on myosin V. Interestingly, while the joint constraint considerably alters the diffusive search space, it has relatively small influence on the myosin V step size and force response. A free diffusion model, for instance, produces similar kinetic behaviors. Our model allows us to address questions related to the biological significance of the joint constraint. We find, for example, that the constraint does not necessarily speed up the binding time, despite narrowing the space of the diffusive search. However it does narrow the width of the forward step distributions on actin, allowing the head to more consistently target the actin binding sites at half-helical length intervals. Finally, the model provides testable predictions of new quantities yet to be probed by experiments, including the stomp distribution near the stall regime and the robust run length under off-axis forces.

## Results

### Theoretical model for myosin V dynamics

In the following sections we describe the main features of our polymer structural model for myosin V, the actin filament geometry, and how the model allows us to predict diffusion and stepping behavior, including the probabilities of various kinetic pathways. The full mathematical details of the analytical theory can be found in Appendix 1. The details of the BD simulations, which we used to validate the analytical theory results, are described in Appendix 2.

### Polymer model and actin filament geometry

Following *Hinczewski et al. (2013)*, we model the actin-binding heads and lever arm domains of myosin V as semi-flexible polymer chains with length $L$ and persistence length $l_p$. We will denote each polymer chain as a 'leg' of the motor (consisting of the combined head and lever arm domains), and refer to trailing or leading legs depending on whether a given leg is further away from or closer to the barbed (plus) end of actin, respectively. Several experiments measuring myosin step distributions and other properties have been carried out comparing wild-type myosin V to mutants where the lever arm length is altered through addition or deletion of IQ motifs (*Yildiz et al., 2003*; *Sakamoto et al., 2005*; *Oke et al., 2010*). We assume the actin-binding head and IQ motifs are each approximately 5 nm in length, so that wild-type myosin V (with 6IQ motifs) has $L = 35$ nm and the 4IQ and 8IQ mutants have $L = 25$ nm and $L = 45$ nm respectively. The persistence length has estimated values ranging from $l_p \approx 100$ nm (*Howard and Spudich, 1996*) to $l_p \approx 375$ nm (*Vilfan, 2005a*). While fitting our model to experimental data we allow the persistence length to vary within this range, though the model predictions are qualitatively similar for any $l_p \gg L$ in the stiff leg regime.

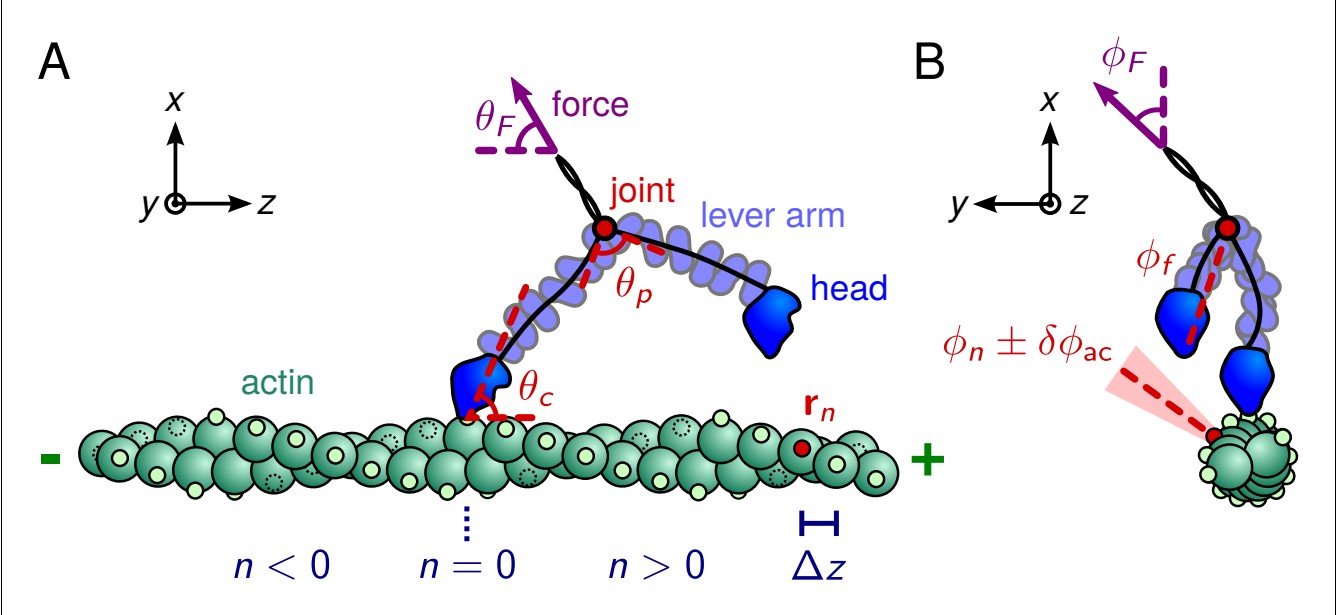

**Figure 1.** Myosin V geometry. (**A**) Side view, with the actin filament plus end oriented toward the $\hat{z}$ direction. Small circles on the actin monomers denote the binding sites $\mathbf{r}_n$, described by *Equation 1*. The site $n = 0$ corresponds to the position of the bound head. The bound polymer leg has a preferred post-power stroke direction in the $x - z$ plane defined by a constraint angle $\theta_c$ relative to the $\hat{z}$ axis. Due to the hypothesized structural constraint at the joint, the preferred angle between the lever arms is $\theta_p$. The force transmitted through the tail domain has a polar angle $\theta_F$ relative to the $-\hat{z}$ direction. (**B**) Front view, with the actin plus end pointing out of the page. Each binding site has an associated outward pointing normal direction with azimuthal angle $\phi_n$. As an example, one such angle is shown for the red-colored site. All azimuthal angles are measured counter-clockwise with respect to the $\hat{x}$ direction. For binding to occur, the head has to be in the vicinity of the site, and oriented approximately along the normal. We approximately capture this condition by a binding criterion that requires the azimuthal angle of the free leg, $\phi_f$, to be anti-parallel to $\phi_n$ within a cutoff range $\pm \delta \phi_{ac}$, highlighted in light red. The load force may have an off-axis component with azimuthal angle $\phi_F$.

The two polymer legs are connected at a joint forming the myosin V dimer (see *Figure 1* for an illustration of the geometry). Recent experimental evidence suggests this this joint does not not rotate freely, but instead has a preferred joint angle $\theta_p$ giving rise to constrained diffusion (*Takagi et al., 2014*; *Andrecka et al., 2015*). In our model, this preferred angle is enforced by a potential $\mathcal{H}_J = \mu_c k_B T[1 - \cos(\theta_J - \theta_p)]$, which is minimum when the inter-leg angle at the joint $\theta_J$ is equal to the preferred angle. The parameter $\mu_c$ is the constraint strength, designating the energy cost to deviations from $\theta_p$, while $k_B$ is Boltzmann's constant and T is temperature. Note that for small angle differences $\theta_J - \theta_p$, the constraint potential $\mathcal{H}_J \approx \frac{1}{2}\mu_c k_B T(\theta_J - \theta_p)^2$ is approximately harmonic. In the limit $\mu_c \to 0$ the joint becomes freely rotating with no preferred angle, the case considered previously *Hinczewski et al. (2013)*. This cosine potential is used throughout the paper, but below we briefly discuss how the form of the inter-leg potential affects the diffusion of the free head. The tail domain, which attaches to cargo, transmits a load force $\mathbf{F}$ to the joint. The direction of the force is parameterized by $\theta_{\mathrm{F}}$, measured clockwise from the $-\hat{\mathbf{z}}$ axis, and $\phi_{\mathrm{F}}$, measured counterclockwise from the $\hat{\mathbf{x}}$ axis. Our main focus is on the behavior of myosin V under zero force and backward force ($\theta_{\mathrm{F}} = \phi_{\mathrm{F}} = 0$), but we also consider off-axis forces at the end of the *Results* section.

The actin-binding heads, located at the ends of the polymer legs in our model, can bind to various sites along the double-helical filamentous actin structure. Actin is composed of two filaments each containing 13 actin subunits per helical rotation, with one binding site per subunit. The filaments run parallel to the $\hat{\mathbf{z}}$ axis, leading to a geometry in which the binding sites $\mathbf{r}_n$, $n = 0, \pm 1, \pm 2, \ldots$ have positions

$$\mathbf{r}_n = R(\cos\phi_n - 1)\,\hat{\mathbf{x}} + R\sin\phi_n\,\hat{\mathbf{y}} + (n/2)\Delta z\,\hat{\mathbf{z}}, \tag{1}$$

where $R = 5.5$ nm is the radius of the helix, $\Delta z = 72/13 \approx 5.5$ nm is the size of each actin subunit, and $\phi_n = -12\pi n/13$ is the angle between adjacent subunits (*Lan and Sun, 2006*). *Equation 1* and all the other key analytical quantities in our theory are summarized in *Table 1* for ease of reference. Even and odd $n$ respectively correspond to subunits on the 1$^{\text{st}}$ and 2$^{\text{nd}}$ filaments of the double helix. Wild-type myosin V steps most frequently to the half-helical sites $n = \pm 13$ located at $z = \pm 36$ nm, while mutants favor other sites depending on their lever arm length (*Sakamoto et al., 2005*; *Oke et al., 2010*).

When myosin V is bound to an actin filament, the lever arm can be in two orientations. After the leg has bound, but before the power stroke has been executed, the lever arm points toward the pointed (minus) end of the actin. After the power stroke, the leg rotates toward the barbed (plus)

**Table 1.** Summary of main analytical results.

| Quantity | Meaning | Definition |
|---|---|---|
| $\mathbf{r}_n$ | position of actin subunits | *Equation 1* |
| $t_{\mathrm{fp}}^n$ | first passage time to subunit $n$ | *Equation 3* |
| $\mathcal{P}(\mathbf{r})$ | equilibrium distribution of the free head position | following *Equation 3* |
| $\mathcal{P}_T^n$ | binding probabilities for trailing leg | *Equation 4* |
| $\mathcal{P}_L^n$ | binding probabilities for leading leg | following *Equation 4* |
| $\mathcal{P}_{\mathrm{dist}}^n$ | distribution of head-to-head distances | *Equation 5* |
| $\mathcal{P}_T(z_n)$ | convolved trailing leg step distribution | following *Equation 5* |
| $\mathcal{P}_L(z_n)$ | convolved leading leg step distribution | preceding *Equation 6* |
| $\mathcal{P}(z_n)$ | full convolved step distribution | *Equation 6* |
| $\hat{\boldsymbol{\mu}}_c'$ | constraint direction (under force) | *Equation 7* |
| $\mathcal{T}'$ | power stroke effectiveness (under force) | *Equation 7* |
| $\mathcal{P}_b/\mathcal{P}_f$ | backward-to-backward step ratio | Step ratio section |
| $z_{\mathrm{run}}$ | mean run length | *Equation 10* |
| $v_{\mathrm{run}}$ | mean run velocity | preceding *Equation 11* |
| $t_{\mathrm{run}}$ | mean run time | *Equation 11* |

end of the actin and is held at an angle $\theta_c$ above the actin. Similar to the inter-leg constraint described above, in our model the preferred forward tilting angle is enacted through a harmonic potential, $\mathcal{H}_c = \frac{1}{2}\nu_c k_B T(\hat{\mathbf{u}}_0 - \hat{\mathbf{u}}_c)^2$. The constraint has strength $\nu_c$, the unit vector $\hat{\mathbf{u}}_0$ is the tangent to the bound leg at the binding point, and the unit vector $\hat{\mathbf{u}}_c$ defines the preferred direction, which lies in the $\hat{\mathbf{x}} - \hat{\mathbf{z}}$ plane at an angle $\theta_c$ from the $\hat{\mathbf{z}}$ axis. As we will see below, steps and stomps occur only when the bound leg is in the post-power stroke orientation. For the purposes of modeling these aspects of the dynamics we do not need to consider a separate constraint potential for diffusion while the bound leg has the pre-power stroke orientation.

Volume exclusion effects introduce another constraint on the orientation of the myosin legs during diffusion. For instance, the myosin heads are unlikely to be found in close proximity due to steric repulsion. Such exclusion interactions apply not only to the myosin heads, but also to the legs, which have to be brought close together in order to accommodate small head separations. Though these interactions can't be explicitly included in the coarse-grained polymer model, we capture the effective repulsion between the myosin heads using the potential $\mathcal{H}_V = k_B T(d_V/r)^6$, where $r$ is the distance between the bound and free myosin heads and $d_V$ is the effective length scale of the repulsion. The magnitude of $d_V$ depends on the details of the interacting legs between the heads, in particular their length $L$. Using BD simulations described below (which explicitly include all volume exclusion interactions between the myosin legs), we estimate $d_V \approx 20$, 27.5, and 35 nm for 4IQ, 6IQ, and 8IQ myosin respectively.

## Brownian Dynamics simulations

We also study the stepping dynamics of myosin V using BD simulations. In the simulations, myosin V is treated as two connected and interacting polymer chains. The Hamiltonian that prescribes the interactions and the resulting structural features like the persistence length, is outlined in Appendix 2. A BD trajectory begins with the trailing lever arm unbinding from actin (mimicking the effect of ATP binding). A power stroke moves the unbound head quickly forward, after which it executes a diffusive search until it finds an energetically favorable actin binding site. As in the analytical model, the actin binding sites are given by *Equation 1*. We trace the diffusive search and record the binding locations. The reported data is averaged over 200 randomized trajectories. The simulations allow us to include the finite sizes of the lever arms, the effect of the volume excluded by actin, and a glass cover-slip that is used in certain experimental setups. Additionally we can test different functional forms for the binding criterion and power stroke execution.

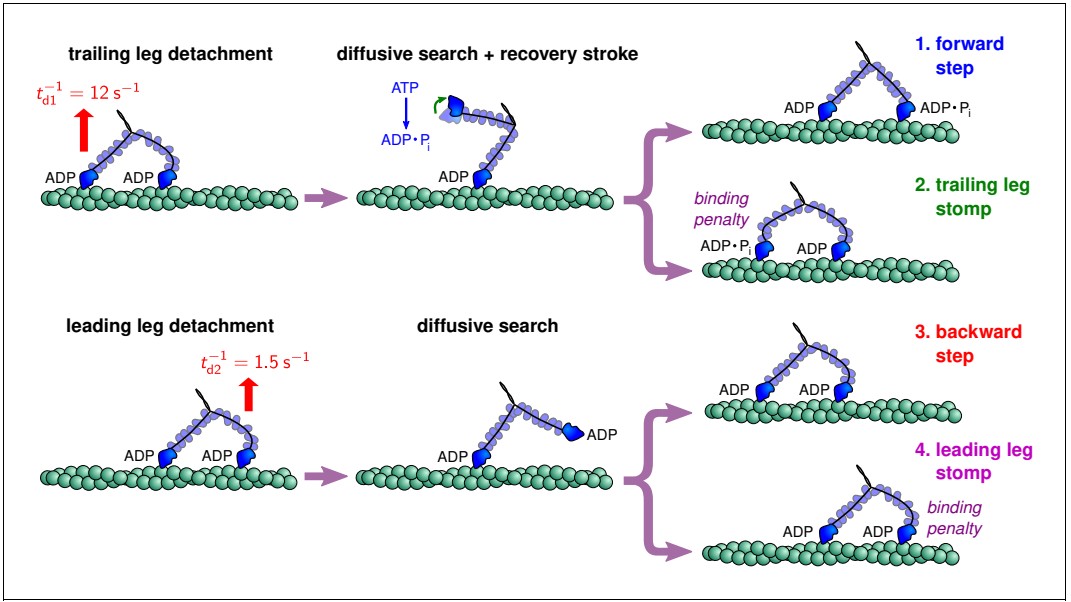

**Figure 2.** Myosin V kinetic pathways.

## Kinetic pathways

Myosin V can exhibit a wide range of behaviors, shown schematically in *Figure 2*, including steps and stomps involving each polymer leg. Before following a particular kinetic pathway, Myosin V starts in a waiting state where both legs are strongly bound to the actin and each head has an associated ADP molecule. Each leg is in the post-power stroke orientation, pointed toward the plus end of the actin, and the leading leg is bent backward under tension. The motor then goes through one of many kinetic pathways, which can be broken into the five categories described below.

## Forward steps

The myosin sits in the waiting state until the ADP molecule unbinds from the trailing head, allowing an ATP molecule to take its place. With the bound ATP, the trailing leg is only weakly associated to the actin and therefore quickly detaches. We assume (except for modeling one run velocity experiment described below) that the ATP concentration is near saturation levels, so that ATP binding occurs rapidly. Then the detachment process occurs roughly at the rate of ADP dissociation $t_{d1}^{-1} = 12\ \mathrm{s}^{-1}$, which has been measured experimentally (*De La Cruz et al., 1999*). Following detachment of the trailing leg, the leading leg, previously under tension, undergoes rapid diffusive relaxation: since the leading leg had been bent backwards, away from its preferred post-power stroke orientation, the leg relaxes toward the preferred orientation, thus leading to the entire system swinging forward toward the plus end of actin (*Hinczewski et al., 2013*; *Dunn and Spudich, 2007*). The timescale $t_r$ of this relaxation depends on the load, but is generally $\leq 5\ \mu s$, based on both theoretical and numerical considerations *Hinczewski et al. (2013)*. Over much longer timescales, the free leg diffusively searches until it reaches a binding site on the actin. If we let the bound leg be attached at the origin $\mathbf{r} = 0$ then the available binding sites are located at $\mathbf{r}_n$ given by *Equation 1*, and forward steps occur when the free leg reaches a site with $n > 0$, assuming the following additional binding criteria are also fulfilled.

Before the detached head can bind to actin, the ATP must hydrolyze, ATP $\rightarrow$ ADP + $\mathrm{P_i}$, which has two primary functions. First, during hydrolysis the free head undergoes the recovery stroke, rotating the head into the pre-power stroke orientation, so that binding to forward sites along the actin is conformationally favorable. Second, the hydrolysis produces an ADP molecule, which is required for the head to strongly associate with actin and successfully bind. The ATP hydrolysis occurs at rate $t_h^{-1} = 750\ \mathrm{s}^{-1}$ (*De La Cruz et al., 1999*) and we assume the reverse reaction rate is negligible.

Once ATP hydrolysis is completed, the free head must diffuse close enough to the actin to appreciably interact and bind. We assume that binding occurs when the head is within a distance $a$ from an actin binding site. An upper bound for the capture radius $a$ is the Debye screening length, which under physiological conditions (KCL concentrations of $25 - 400$ mM) is $\lambda_D \approx 1.9 - 0.5$ nm (*Craig and Linke, 2009*). On these length scales, the detached head can interact with the binding site and we expect near certain binding at slightly shorter distances. Below we find that for $a = 0.4$ nm the model produces quantitative agreement with experimental data. Finally, in addition to requiring close proximity of the free head to an actin binding site, the detached myosin head must also be in the correct orientation for binding to occur (the full details of this binding criterion are described in the section 'First Passage Times and the Binding Acceptance Region' below.)

## Trailing leg stomps

This kinetic pathway starts identically to the forward steps described above, namely the trailing leg dissociation is followed by ATP binding and hydrolysis. In this case, however, the diffusive search finds a site $\mathbf{r}_n$ with $n < 0$. Instead of executing a hand-over-hand step, the myosin stomps and the heads remain in the same relative order on the actin, with a comparably small change in the center of mass position of the myosin. As before, we assume the binding occurs once the head is within a distance $a$ from a binding site with the free leg in a sufficiently acceptable orientation.

The recovery stroke is executed during ATP hydrolysis and before binding, so that the pre-power stroke orientation of the head favors binding to forward sites. After a forward step, the lever arm points toward the minus end of the actin in a relaxed pre-power stroke state. To bind to sites behind the bound leg, however, the free lever arm must bend and point toward the plus end of the actin despite its pre-power stroke orientation with respect to the head. This unnatural configuration puts

the free leg under additional strain, so that there is an energy barrier to binding at backward actin sites. We model this effective barrier by reducing the probability of binding when the free head reaches a radius $a$ of actin sites with $n<0$. Instead of binding with certainty, the myosin head binds with probability $b<1$, which we call the binding penalty. The reduced binding probability, a consequence of the recovery stroke, increases the relative probability that the myosin will step forward, contributing to the biased motion along the actin filament.

## Backward steps

The other possible kinetic pathways start with the leading head detaching from the actin, in contrast to the trailing head detachment which initiates forward steps and trailing leg stomps. After the power stroke, the leading lever arm is under considerable backward tension. This tension dramatically suppresses the release of ADP from the leading head by a factor of 50–70 (*Kodera et al., 2010*; *Rosenfeld and Sweeney, 2004*). Since these events are very rare, an alternative detachment pathway is dominant: the leading head dissociates from the actin but retains the associated ADP molecule (*Kodera et al., 2010*; *Purcell et al., 2005*). This detachment pathway occurs at a slower rate than trailing head detachment, $t_{d2}^{-1} = (g t_{d1})^{-1}$, where $g>1$ is the gating ratio. Under backward force of ~2 pN, single-headed myosin V has been observed to detach from actin at a rate of 1.5 s$^{-1}$ (*Purcell et al., 2005*), while previous theoretical calculations using the above described polymer model estimate the backward force on the leading leg to be 2.7 pN (*Hinczewski et al., 2013*) when myosin V sits in the waiting state with both legs bound. We therefore use $t_{d2}^{-1} = 1.5$ s$^{-1}$, which corresponds to the gating ratio $g = 8$.

Once the leading head unbinds from the actin, it executes a diffusive search until it reaches an actin binding site. Note that the bound leg is in the post-power stroke state, so the diffusion is statistically identical to that which occurs during forward steps and trailing leg stomps. In this case, however, the free head has an associated ADP molecule immediately after detachment, so the free leg can rebind without undergoing ATP hydrolysis. Furthermore, because ATP hydrolysis is not executed, the free head remains in the post-power stroke state orientation which favors binding to backward sites. If we let the bound leg be at the origin $\mathbf{r} = 0$, a backward step occurs when the diffusive search finds an actin site $\mathbf{r}_n$ with $n<0$. Hence, the free head orientation favors binding to these sites and binding occurs with probability 1 when the head diffuses within a distance $a$ of the target site and the free leg is within the acceptance region.

## Leading leg stomps

Similar to the trailing leg stomps described above, the myosin V can also execute a leading leg stomp (*Kodera et al., 2010*; *Andrecka et al., 2015*). This kinetic pathway begins with leading leg detachment, identical to the backward step. To perform a stomp, the free head diffuses to an actin binding site $\mathbf{r}_n$ with $n>0$. Since the free head is in the pre-power stroke state, binding to these forward sites requires an unnatural deformation of the free leg, which as in the case of trailing leg stomps, introduces an effective energy barrier to these events. Therefore, when the free head diffuses within a radius $a$ of a target site with the free leg in the acceptance region, binding occurs with probability $b<1$, given by the binding penalty.

## Termination

The final kinetic pathway occurs when the bound myosin head detaches before the free head finds an open actin binding site, terminating the processive run of myosin V. We assume that the bound leg has a constant detachment rate equal to $t_{d1}^{-1}$, independent of the backward load force exerted on the motor.

These five pathways complete the kinetic description of myosin V procession. The first four pathways, forward steps, leading and trailing leg stomps, and backward steps, each return the myosin to its waiting state with both legs bound, while termination is a complete dissociation from the actin. Therefore the myosin will continue to execute steps and stomps until termination which ends the run. For notational convenience below, we will denote the binding penalty of a given actin site $\mathbf{r}_n$ following trailing head detachment as $b_n$. Then $b_n = 1$ for $n>0$, there is no penalty for forward steps, and $b_n = b<1$ if $n<0$. The binding penalty following leading head detachment is $b_{-n}$. If we sum the probabilities over all binding sites possible in a given pathway, we get the overall probability of each

type of step and stomp. We define $\mathcal{P}_\mathrm{f}$, $\mathcal{P}_\mathrm{Ls}$, $\mathcal{P}_\mathrm{Ts}$, $\mathcal{P}_\mathrm{b}$, and $\mathcal{P}_\mathrm{t}$ to be the overall probability of myosin executing a forward step, leading/trailing leg stomp, backward step, and termination respectively, starting from the waiting state. The above pathways are dominant when the backward load forces are below or slightly above the stall force, for which experimental estimates range from $\approx 1.9 - 3$ pN (*Mehta et al., 1999*; *Veigel et al., 2002*; *Kad et al., 2008*; *Uemura et al., 2004*; *Cappello et al., 2007*; *Gebhardt et al., 2006*). For large super-stall forces ($\geq 4$ pN), the myosin can experience power stroke reversal, swinging the leading lever arm back toward the minus end of the actin (*Sellers and Veigel, 2010*). At these high forces the diffusive search can therefore occur with the bound leg in the pre-power stroke orientation, so additional pathways must be added to the above model to accurately describe the super-stall behavior of myosin V. We restrict our focus to the sub-stall and stall regimes.

## First passage times and the binding acceptance region

The key quantities necessary for our theoretical description of myosin V processivity are the mean first passage times, $t_\mathrm{fp}^n$, of the free head to each of the actin binding sites $\mathbf{r}_n$. These determine the relative binding probabilities for each of the many possible target actin sites. Combined with the above described step and stomp pathways, this gives us a complete kinetic description of myosin V informed by the polymer nature of the lever arms. Below we use the first passage times to compute various experimentally observable quantities including step distributions, the forward/backward step ratio, and the mean run length and velocity.

Our polymer model admits an accurate approximate analytical expression for the mean first passage time. In past theoretical work by *Hinczewski et al. (2013)*, it was shown that the mean first passage time is approximately

$$t_\mathrm{fp}^n \approx \frac{1}{4\pi D_\mathrm{h} a \mathcal{P}(\mathbf{r}_n)},\tag{2}$$

where $a$ is the capture radius, $\mathcal{P}(\mathbf{r})$ is the probability density of the free head being at position $\mathbf{r}$, and $D_\mathrm{h} = 5.7 \times 10^{-7}$ cm²/s is the diffusion constant of the myosin V head which was estimated using the program HYDROPRO (*Ortega et al., 2011*) with the Protein Data Bank structure 1W8J (*Coureux et al., 2004*). Here $\mathcal{P}(\mathbf{r})$ is the equilibrium distribution achieved at timescales greater than $t_r$, the relaxation timescale of the polymer. This result relies on the separation of timescales between the polymer relaxation of the lever arms and the first passage of the diffusive search. Previous BD simulations (*Hinczewski et al., 2013*) found that the lever arm relaxation timescale is $t_\mathrm{r} \lesssim 5$ µs, which is two orders of magnitude smaller than the smallest mean first passage times, $t_\mathrm{fp}^\mathrm{min} \sim \mathcal{O}(0.1\mathrm{ms})$. Another requirement is that the time to diffuse the distance $a$, $t_a = a^2/D_\mathrm{h} \approx 2.8$ ns be much smaller than the relaxation time $t_\mathrm{r}$. This condition, which is clearly satisfied since $t_a/t_r \approx 5 \times 10^{-4} \ll 1$, guarantees that after the free head reaches the capture radius it can undergo fast microscopic rearrangements required to bind without significant conformational changes in the rest of the lever arm.

In our model there is an additional condition necessary for binding: the free leg must be within the angular acceptance region with respect to the actin subunit. Therefore, we are actually interested in the mean first passage time to finding a binding site and simultaneously having the correct orientation. Electron and atomic force microscopy imaging of myosin V indicate that while bound the myosin head attaches approximately perpendicular to the outer surface of an actin subunit (*Oke et al., 2010*; *Kodera et al., 2010*). This binding, which involves the interaction of a specific region of the head with a corresponding region on the actin subunit, is mimicked in our coarse-grained model through an angular criterion: we require that the angle between the free leg and the outward pointing normal ($\hat{\mathbf{r}}_n \times \hat{\mathbf{z}}$) at the target actin site be smaller than $\delta\phi_\mathrm{ac}$, which defines a conical acceptance region in which binding is allowed. Based on fits to experiments described below, we set $\delta\phi_\mathrm{ac} = 55.6°$. We assume binding occurs with probability 1 when the free head is within a distance $a$ from a binding site and the free leg is inside the acceptance region.

Though this angular criterion is straightforward to implement in our BD simulations, for the analytical theory it significantly complicates the calculation. Hence we simplify the angular criterion in the analytical model, in a way that preserves most of the physical effect while making the derivation of the mean first passage time tractable. Instead of a conical acceptance region around the outward pointing normal to the target actin site, we only require an azimuthal angle similar to the normal.

This amounts to replacing the probability density $\mathcal{P}(\mathbf{r}_n)$ in *Equation 2* with the joint density of finding the free head at position $\mathbf{r}_n$ and the free leg simultaneously having azimuthal orientation similar to that of the binding site, $\mathcal{P}(\mathbf{r}_n, \delta\phi_1 > (\phi_f + \pi) - \phi_n > -\delta\phi_2)$. Here $\phi_f$ is the azimuthal angle of the end-to-end vector for the free myosin leg and $\phi_n = -12\pi n/13$ is the azimuthal angle of an actin binding site (note the factor of $\pi$ enters because the free leg points toward the binding site). The angles are shown in *Figure 1B*. The acceptance region defined by $(\delta\phi_1, \delta\phi_2)$ will for simplicity be taken as symmetric, $\delta\phi_1 = \delta\phi_2 = \delta\phi_{\mathrm{ac}}$, but we do not rule out the possibility that asymmetry in the myosin head or actin binding pocket favors binding from one direction. Similar effects have been previously observed for myosin V, for instance the suppression of ADP dissociation and head detachment depends asymmetrically on the direction of off-axis forces (*Oguchi et al., 2010*). The acceptance region requirement increases the first passage time, because in some cases the free head will diffuse to a binding site but have the wrong orientation and fail to bind. Therefore, the required separation of timescales still holds and, with a symmetric acceptance region, the mean first passage time is,

$$t^n_{\mathrm{fp}} \approx \frac{1}{4\pi a D_{\mathrm{h}} \mathcal{P}(\mathbf{r}_n, \delta\phi_{\mathrm{ac}} > |(\phi_f + \pi) - \phi_n|)}. \tag{3}$$

The use of the simplified angular criterion in the analytical theory gives results similar to the BD simulations, where the full conical acceptance region was used.

The properties of the polymer legs as well as the influence of the binding acceptance conditions are implemented through the joint distribution $\mathcal{P}(\mathbf{r}_n, \delta\phi_{\mathrm{ac}} > |(\phi_f + \pi) - \phi_n|)$. Adapting a polymer mean field theory (*Hinczewski et al., 2013*; *Thirumalai and Ha, 1998*) and exploiting the fact that the myosin lever arms are in the stiff regime $l_p \gg L$, we derive an approximate analytical expression for this distribution. We also derive an expression for $\mathcal{P}(\mathbf{r})$ at any point $\mathbf{r}$ in space, which describes the equilibrium distribution of the free leg during diffusion and captures the effects of the inter-leg structural constraint, the bound leg constraint, and the load force (Appendix 1). Assuming the binding events along with trailing/leading head detachment and hydrolysis are each Poisson processes with rates $(t^n_{\mathrm{fp}})^{-1}$, $t^{-1}_{\mathrm{d1}}$, $t^{-1}_{\mathrm{d2}}$, and $t^{-1}_{\mathrm{h}}$ respectively, we derive the probabilities of passage through each kinetic pathway and analytical expressions for observable quantities including step distributions as well as mean run length and velocity.

## Estimating the joint constraint strength from the free head spatial distribution during stepping

Until recently, the joint between the myosin V lever arms was believed to be freely rotating, allowing the detached head to perform a full three-dimensional diffusive random walk while searching for actin binding sites. This hypothesis was supported by the work of *Dunn and Spudich (2007)*, who used darkfield microscopy to image (from above) the diffusive path of 40 nm gold nanoparticles attached to the leg of myosin V. Their measured diffusion contours are radially symmetric about the inter-leg joint implying a freely rotating hinge. Other experiments also suggest a freely rotating joint (*Shiroguchi and Kinosita, 2007*; *Fujita et al., 2012*; *Beausang et al., 2013*), while theoretical and numerical models based on the free diffusion hypothesis largely agree with experimental data (*Hinczewski et al., 2013*; *Craig and Linke, 2009*; *Mukherjee et al., 2017*). Despite this body of evidence, recent experiments with improved spatiotemporal imaging resolution indicate that the myosin V joint might not freely rotate, instead preferring a particular inter-leg angle due to structural constraints. In particular, electron micrographs of *Takagi et al. (2014)* show that in the absence of actin, myosin V has a weakly preferred inter-leg angle of approximately 110°. Further, by tracking 20 nm gold particles attached to the N-terminus of the myosin head with interferometric scattering microscopy, *Andrecka et al. (2015)* observed a multi-peaked diffusion contour, contradicting the presence of a freely rotating joint.

Our polymer model accounts for structural constraints on diffusion encoded through the inter-leg angle preference $\theta_p$ and constraint strength $\mu_c$. We let $\theta_p = 83.0°$, as determined by fitting to step distribution and force dependence experiments discussed below. While this value is somewhat smaller that the actin-free *Takagi et al. (2014)* observations, we do not expect an exact correspondence. In particular, the observed angle will be larger than $\theta_p$ due to volume exclusion interactions and the effective preferred angle in the presence of actin may be further altered through the influence of the bound leg on the overall conformation of the protein. The presence of a joint constraint

changes the diffusion of the free leg so that it swings through a nearly one-dimensional arc while searching for binding sites rather than exploring the full three-dimensional space. In order to visualize the diffusion, we use the probability density of the location of the free head $\mathcal{P}(\mathbf{r})$ projected onto the two dimensional $z-x$, $z-y$, and $y-x$ planes as well as the cylindrical plane $z-\rho$, where $\rho = x^2 + y^2$. The projection is performed by integrating over the remaining degree of freedom, for instance $\mathcal{P}(x, z) = \int dy \mathcal{P}(\mathbf{r})$.

The diffusion contour measured by *Andrecka et al. (2015)* and shown in *Figure 3I*, corresponds to the $z-x$ projection since they imaged through a glass surface parallel to both the myosin bound leg and direction of motion. We evaluated this contour using our analytical theory for several values of the inter-leg constraint strength, $\mu_c = 0, 3, 5,$ and 12 (*Figure 3A–D*). These calculations agree well with diffusion contours measured from Brownian dynamics simulations for the same values of $\mu_c$ (*Figure 3E–H*). Our results show that a non-zero inter-leg constraint $\mu_c > 0$ is required to produce a multi-peaked contour similar to that measured by *Andrecka et al. (2015)*. In particular, when $\mu_c = 5$ the predicted diffusion agrees well with the experimental data, while $\mu_c = 3$ and $\mu_c = 12$ respectively produce contours with lower and higher peaks than the measured distribution. Therefore, we estimate the energy of the inter-leg constraint lies within this range $3K_BT \lesssim \mu_c K_B K \lesssim 12K_BT$. Incidentally,

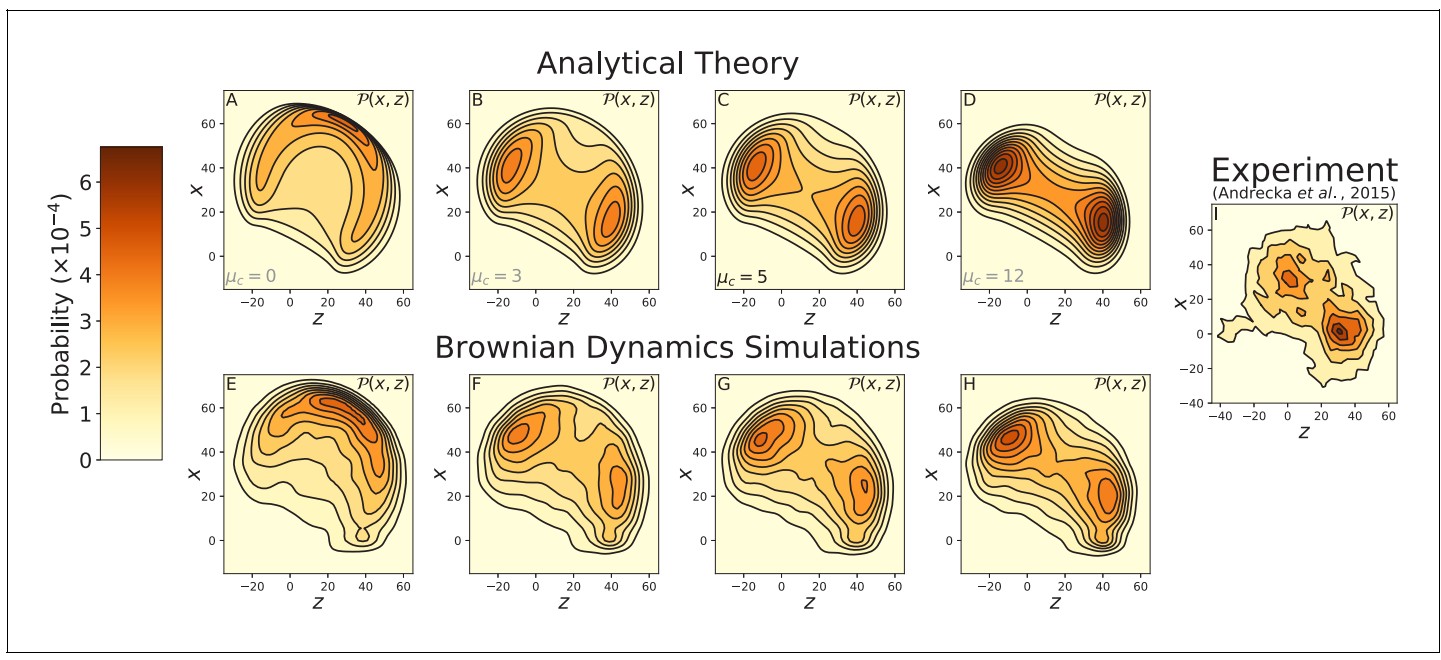

**Figure 3.** Contours of the myosin V free head position distribution $\mathcal{P}(\mathbf{r})$ projected onto the $z-x$ plane. Top row: theoretical predictions for (**A**) free diffusion ($\mu_c = 0$) and (**B–D**) constrained diffusion with inter-leg constraint strength (**B**) $\mu_c = 3$, (**C**) $\mu_c = 5$, and (**D**) $\mu_c = 12$. Bottom row: the corresponding contours measured from Brownian dynamics simulations, with inter-leg constraint strength (**E**) $\mu_c = 0$, (**F**) $\mu_c = 3$, (**G**) $\mu_c = 5$, and (**H**) $\mu_c = 12$. (**I**) Experimental measurements of the diffusion by *Andrecka et al. (2015)*. Adding an inter-leg constraint potential produces a multi-peaked diffusion pattern. The heights of the peaks are similar to the experimental measurements for $3 \lesssim \mu_c \lesssim 12$. Note that the $x = 0$ axis in the experimental data corresponds to the position of the gold bead attached to the myosin head when the head is bound to actin. Given the ~5 nm size of the head and ~10 nm radius of the bead, this accounts for the approximately 15 nm vertical shift between the theoretical/simulation distributions and experiment. In the former the $x = 0$ axis corresponds to the top of the actin filament (where the bound head is attached).

The online version of this article includes the following figure supplement(s) for figure 3:

**Figure supplement 1.** Alternative projections of the constrained diffusion ($\mu_c = 5$).

**Figure supplement 2.** Alternative projections of the free diffusion ($\mu_c = 0$).

**Figure supplement 3.** Constrained diffusion of myosin V under 2 pN backward force.

**Figure supplement 4.** Constrained diffusion of the 4IQ myosin mutant.

**Figure supplement 5.** Constrained diffusion of the 8IQ myosin mutant.

**Figure supplement 6.** Effect of the form of the joint potential on free head diffusion.

**Figure supplement 7.** Effects of cover-slip volume exclusion on diffusion.

the upper bound 12 k$_B$T is also approximately the energy at which our model can no longer accurately predict the multi-peaked step distribution of 8IQ mutants discussed in the next section.

For the cases where the constraint is strong enough to induce a bimodal distribution, the two peaks are roughly equal in height both in the analytical theory and in simulations. In constrast, the experimental distribution is asymmetric, with the peak closer to the actin about 1.5 times higher than the one further away from the actin. While we cannot rule out that some aspect of this asymmetry may be due to additional structural features not present in the model, it is also likely to be at least in part an artifact of the finite time resolution of the experiment and possibly consequences of attaching a gold nanoparticle to infer the motility of the detached motor head. The peak region further away from actin is the one that is initially visited after detachment of the trailing head. If one cannot resolve the precise moment of detachment, one might not be able to fully capture this portion of the distribution in the experimental data. In fact, *Andrecka et al. (2015)* measured the distribution for a range of ATP concentrations, and as the ATP was progressively increased from 1μM to 1 mM, the height of the far peak decreased relative to the near peak, even though the distribution should in principle be independent of ATP. Detachment is much more rapid at high ATP, however, and the resulting practical difficulties in collecting trajectories were mentioned in the paper.

The other projections complete the picture of the free head diffusion. These are shown for the constrained diffusion model ($\mu_c = 5$) in *Figure 3—figure supplement 1* and for the free diffusion model ($\mu_c = 0$) in *Figure 3—figure supplement 2*. When the inter-leg constraint is present, the $y - x$ (front view) and $z - y$ (top view) projections together show that during its diffusive search the free leg swings out away from the actin axis at a particular height, with motion similar to that of a drawing compass. We also computed and simulated diffusion contours for myosin V under 2 pN backward force (*Figure 3—figure supplement 3*) as well as for the 4IQ ((*Figure 3—figure supplement 4*) and 8IQ mutants (*Figure 3—figure supplement 5*). Under force, the free head distribution rotates toward the minus end of the actin changing which target binding sites are most likely to be found during a step. The diffusion of mutants is almost identical to wild-type myosin V, but is scaled up or down based on lever arm length. This suggests that qualitative differences in stepping between mutants (discussed below) are due entirely to actin geometry.

Our calculation of the free head spatial distribution allows us to test how different forms of the inter-leg potential affect the diffusive step of myosin V. *Figure 3—figure supplement 6* shows the Kullback-Leibler (KL) divergence $D_{KL}(\mathcal{P}_{\cos}|\mathcal{P}) = \int d^3\mathbf{r}\mathcal{P}_{\cos}(\mathbf{r}) \log[\mathcal{P}_{\cos}(\mathbf{r})/\mathcal{P}(\mathbf{r})]$ between the free head distribution $\mathcal{P}_{\cos}(\mathbf{r})$ arising from the cosine joint potential $\mathcal{H}_J = \mu_c k_B T[1 - \cos(\theta_J - \theta_p)]$ introduced above, and the distribution $\mathcal{P}(\mathbf{r})$ arising from a general quartic inter-leg potential $\mathcal{H}_J = \mu_c k_B T[(\Delta\theta)^2/2 + h_3(\Delta\theta)^3/3! + h_4(\Delta\theta)^4/4!]$, where $\Delta\theta = \theta_J - \theta_p$. Also shown in this figure are the $x - z$ diffusion contours for a few representative alternative potentials. This calculation provides a quantitative picture of how much the diffusion changes over a range of joint potentials. Specifically, our results are insensitive to the form of the potential when the anharmonic terms ($h_3$, $h_4$, etc.) are small enough to not introduce a new energy minimum in the physically relevant range of inter-leg joint angles ($\Delta\theta \lesssim \pi/2$). In this case, the resulting diffusion contour is bimodal and qualitatively similar to that shown in *Figure 3*.

The imaging techniques used by *Andrecka et al. (2015)* require a glass cover-slip which excludes half the space available for the myosin diffusive search. They report no evidence of interactions between the myosin and the surface. However, entropic forces due to volume exclusion do influence the diffusion. While it is unlikely these effects would induce a multi-peaked diffusion contour, we explicitly check this by adding a potential barrier excluding half of space in Brownian dynamics simulations (see *Figure 3—figure supplement 7*). Our results rule out the appearance of a multi-peaked distribution due solely to volume exclusion and confirm the cover-slip only slightly changes the shape of the diffusion contours.

The discrepancy between the single-peaked distribution of the *Dunn and Spudich (2007)* experiment and the the multi-peaked distribution of *Andrecka et al. (2015)* remains a controversy, but as noted in the latter work it can be partially resolved by adding localization noise and re-binning their data, accounting for the differences in gold nanoparticle size and the associated measurement precision between the two experiments. The fact that *Dunn and Spudich (2007)* attached their gold nanoparticle labels on the lever arm closer to the joint may also have contributed to concealing the constrained diffusion: the closer the label is to the joint, the closer together the two peaks appear in

the distribution, and the harder it becomes to distinguish one from two peaks given finite experimental spatial resolution.

In the following sections, we use our model to test the constrained diffusion hypothesis against the large body of existing experimental data on myosin V. In the *Discussion*, we compare the constrained and free diffusion models and suggest further experiments to more conclusively discern between these competing myosin V diffusion hypotheses.

## Constrained diffusion model predicts zero force step distributions

The spatial fluctuations of motor proteins, among them distributions of step sizes, have long been an intriguing aspect of their behavior (*Das and Kolomeisky, 2008*). Let us first consider the step size distribution under zero backward force, one of the most commonly measured aspects of myosin V dynamics. The classic experiment by *Yildiz et al. (2003)* measured one of the first myosin step distributions using fluorescence imaging with one nanometer accuracy (FIONA). Later experiments using FIONA (*Sakamoto et al., 2005*) and electron microscopy (*Oke et al., 2010*) determined step distributions for both wild-type myosin as well as 4IQ and 8IQ mutants. The FIONA experiments measure the distance travelled by the trailing head while executing a forward step, while the electron micrographs were used to directly determine the number of actin sites separating legs while bound. These experiments qualitatively agree, with the wild-type distribution peaked at the half-helical site ($z$ = 36 nm) and an approximately linear relation between step size and lever arm length, though *Sakamoto et al. (2005)* measured slightly shorter 4IQ steps than *Oke et al. (2010)*. Furthermore, *Sakamoto et al. (2005)* found that, due to the longer leg length, 8IQ mutants can reach the full helical actin sites ($z$ = 72 nm), giving rise to multi-peaked step distributions. This effect was not observed by *Oke et al. (2010)*, which they note is likely because their image processing technique did not resolve large steps for which the myosin is stretched out very close to the actin.

Our polymer model gives the step distributions in terms of the structural parameters and kinetic rates described in the preceding sections. As shown in Appendix 1, after trailing leg detachment the probability of a step/stomp to site n is

$$\mathcal{P}_{\mathrm{T}}^{n} = \frac{b_n t_{\mathrm{d1}}^2}{t_{\mathrm{fp}}^n (1 + r_{\mathrm{T}} t_{\mathrm{d1}})(t_{\mathrm{d1}} + t_{\mathrm{h}})}, \tag{4}$$

where $r_{\mathrm{T}} = \sum_n b_n (t_{\mathrm{fp}}^n)^{-1}$. The step/stomp probabilities for the leading leg $\mathcal{P}_{\mathrm{L}}^n$ are given by the same expression with the substitutions, $b_n \to b_{-n}$, $t_{\mathrm{h}} \to 0$, and $r_{\mathrm{T}} \to r_{\mathrm{L}} = \sum_n b_{-n}(t_{\mathrm{fp}}^n)^{-1}$ to account for the lack of ATP hydrolysis and recovery stroke in these pathways. The trailing leg detachment occurs with probability $g(1+g)^{-1}$ while the leading leg detachment occurs with probability $(1+g)^{-1}$. Thus, the probability of observing a bound myosin with $n$ subunits between the leading and trailing heads is

$$\mathcal{P}_{\mathrm{dist}}^n = \frac{g}{1+g}(\mathcal{P}_{\mathrm{T}}^n + \mathcal{P}_{\mathrm{T}}^{-n}) + \frac{1}{1+g}(\mathcal{P}_{\mathrm{L}}^n + \mathcal{P}_{\mathrm{L}}^{-n}), \quad n>0. \tag{5}$$

This distribution (normalized to remove termination pathways), describes the probability of finding $n$ actin sites between bound heads, which was measured in the *Oke et al. (2010)* experiment. The overall step sizes, that is the distances traveled by the trailing leg in executing a forward step, have a distribution given by the convolution $\mathcal{P}_{\mathrm{T}}(z_n) = (\mathcal{P}_{\mathrm{dist}} * \mathcal{P}_{\mathrm{T}})[n]$, where $z_n = n\Delta z/2$. The distribution $\mathcal{P}_{\mathrm{T}}(z_n)$ includes both steps and stomps. For comparison to forward step distributions measured in experiments, we exclude the stomps by disallowing binding behind the bound leg and setting $\mathcal{P}_{\mathrm{T}}^n = 0$ for $n<0$ in the convolution. To account for the measurement resolution in FIONA experiments, we convolve this distribution with 1 nm Gaussian noise and bin the data identically to *Sakamoto et al. (2005)*.

To fit step distributions, we fix the binding penalty $b$, capture radius $a$, and power stroke effectiveness defined as $\mathcal{T} = 1 + 20\nu_v/(20 + 7\kappa\nu_c)$, where $\kappa = L/l_p$, determined via fitting to force dependence data (described below). The variable $\mathcal{T}$ measures the energy loaded by the power stroke in the effective spring formed by the lever arms (*Hinczewski et al., 2013*). We vary the bound leg constraint angle $\theta_c$, the acceptance region size $\delta\phi_{\mathrm{ac}}$, and inter-leg preferred angle $\theta_p$ to minimize the Kullback-Leibler (KL) divergence between experimental step distributions and theoretical predictions for 4, 6, and 8 IQ myosin V. We also let the persistence length $l_p$ and bound leg constraint strength

$\nu_c$ vary along curves of constant $\mathcal{T}$, but the model predictions are robust to such parameter variation. Informed by these fits, we choose a set of parameters for which agreement between theory and experiment is excellent for all distributions. We further refine the parameter choices by alternating between fitting step distributions and force dependence behavior, feeding the parameters determined by one fit into the other. We find that $\theta_c = 65.0°$, $\delta\phi_{ac} = 55.6°$ and $\theta_p = 83.0°$ optimizes the agreement between our theory and the full set of experimental step distributions. These and all the other parameter values in the model are summarized in *Table 2*. This constraint angle is similar to that indicated by imaging experiments (*Walker et al., 2000*; *Kodera et al., 2010*; *Lewis et al., 2012*), while the preferred inter-leg angle is slightly smaller than that observed for unbound myosin V (*Takagi et al., 2014*), which, as noted above, could be due to volume exclusion effects and the influence of binding to actin. The azimuthal constraint $\delta\phi_{ac}$ corresponds to an acceptance region of ~111°, which plays a key role in restricting binding too far from the half-helical and full-helical actin sites. Step distributions and direct measurements of the azimuthal trajectory of myosin indicate steps with large azimuthal rotations are rare (*Lewis et al., 2012*), likely due to orientational binding constraints.

The theoretical head separation distributions are shown in *Figure 4*, row 1 alongside data from *Oke et al. (2010)* while in *Figure 4*, row 2 we plot the full convolved step distributions. Since *Oke et al. (2010)* were not able to resolve large head separations for 8IQ mutants, we compute an alternative 'small steps' distribution in which large head separation (> 18 actin subunits) is not considered. The theoretical distribution peaks and width agree well with *Oke et al. (2010)*, especially

**Table 2.** Summary of myosin V model parameters.

For the parameters identified as fit to experiments, the following approach was used: as described in the text, $\theta_c$, $\theta_p$, and $\delta\phi_{ac}$ were varied to fit the step distributions, while $b$, $a$, and $\mathcal{T} = 1 + 20\nu_c/(20 + 7\kappa\nu_c)$ were varied to fit the force response data. Parameters $l_p$ and $\nu_c$ were also allowed to vary along curves of constant $\mathcal{T}$ while fitting the step distributions.

| Parameter | Value | Source |
|---|---|---|
| **Mechanical Parameters** | | |
| Leg contour length, $L$ | 35 nm | *Craig and Linke, 2009* |
| Head diffusivity, $D_h$ | $5.7 \times 10^{-7}$ cm$^2$/s | *Ortega et al., 2011*; *Coureux et al., 2004* |
| Leg persistence length, $l_p$ | 350 nm | Fit to experiment*, *Howard and Spudich, 1996*; *Vilfan, 2005a* |
| Bound leg constraint angle, $\theta_c$ | 65.0° | Fit to experiment*, *Lewis et al., 2012* |
| Bound leg constraint strength, $\nu_c$ | 261 | Fit to experiment |
| Inter-leg preferred angle, $\theta_p$ | 83.0° | Fit to experiment*, *Takagi et al., 2014* |
| Inter-leg constraint strength, $\mu_c$ | 5 | Fit to experiment |
| **Binding Parameters** | | |
| Actin radius, $R$ | 5.5 nm | *Lan and Sun, 2006* |
| Actin monomer size, $\Delta z$ | 72/13 nm | *Lan and Sun, 2006* |
| Actin rotation angles , $\phi_n$ | $-12\pi n/13$ | *Lan and Sun, 2006* |
| Capture radius, $a$ | 0.4 nm | Fit to experiment*, *Craig and Linke, 2009* |
| Binding penalty, $b$ | 0.045 | Fit to experiment |
| Acceptance region, $\delta\phi_{ac}$ | 55.6° | Fit to experiment |
| **Chemical Rates** | | |
| Hydrolysis rate, $t_h^{-1}$ | 750 s$^{-1}$ | *De La Cruz et al., 1999* |
| TH detachment rate†, $t_{d1}^{-1}$ | 12 s$^{-1}$ | *De La Cruz et al., 1999* |
| LH detachement rate, $t_{d2}^{-1}$ | 1.5 s$^{-1}$ | *Purcell et al., 2005* |
| Gating ratio, $g = t_{d2}/t_{d1}$ | 8 | |

*Fits restricted to physically plausible parameter ranges as determined from the indicated literature.

†The TH detachment rate assumes saturating ATP conditions. This is used throughout the paper except for the low ATP run velocity calculation (see Run velocity).

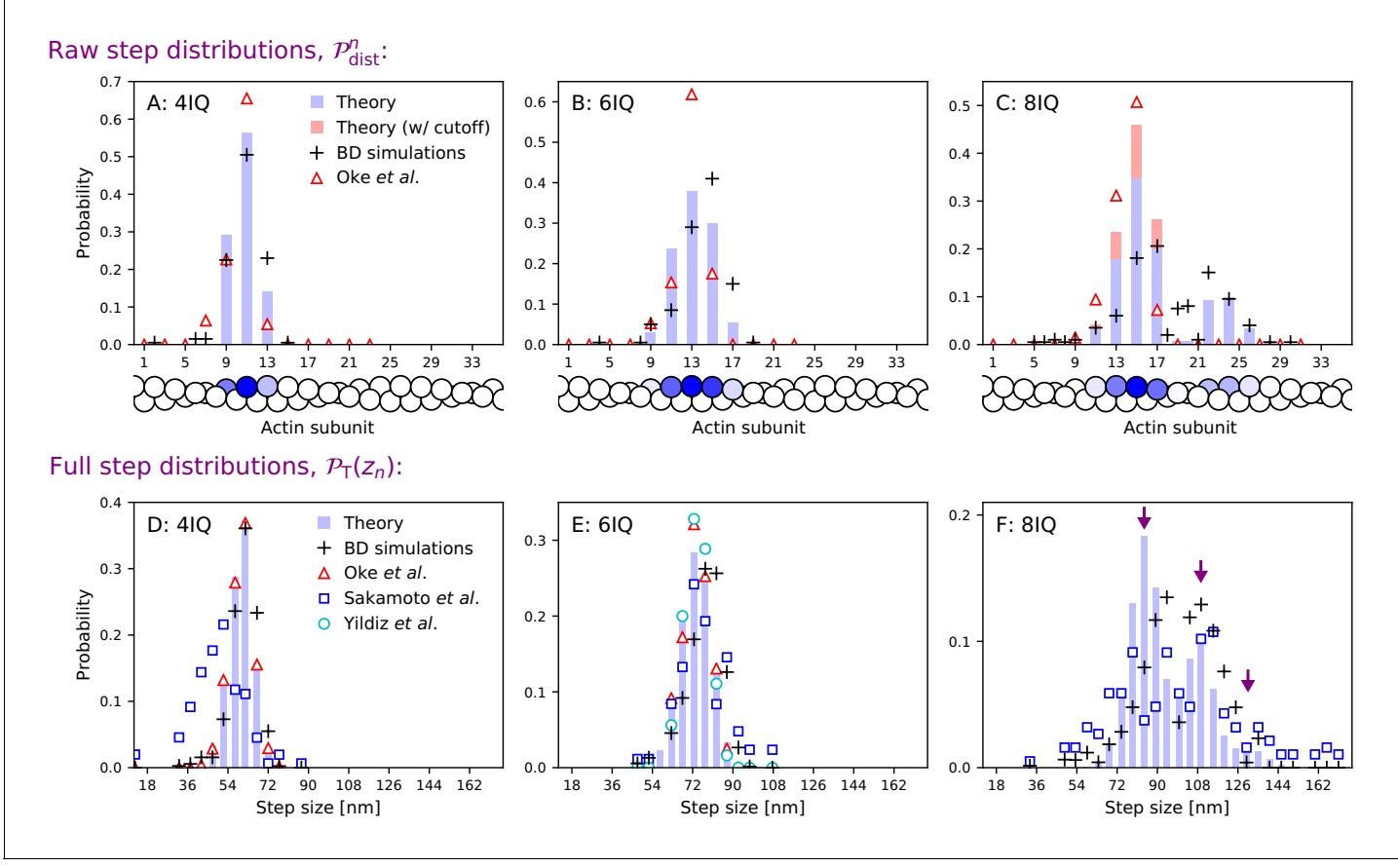

**Figure 4.** Step size distributions for myosin V and mutants with altered leg length. Top row: raw step distributions for (**A**) the 4IQ mutant, (**B**) the 6IQ wild-type, and (**C**) the 8IQ mutant. Bottom row: full (convolved) step distributions for (**D**) the 4IQ mutant, (**E**) the 6IQ wild-type, and (**F**) the 8IQ mutant, with three theoretical peak locations indicated by arrows. Theoretical distributions are shown as histograms with Brownian dynamics simulations and experimental data from *Oke et al. (2010)*, *Sakamoto et al. (2005)*, and *Yildiz et al. (2003)* indicated by symbols. The raw data from *Oke et al. (2010)* is convolved and binned in the bottom row. Since the imaging methods used in this experiment did not resolve large steps taken by the 8IQ mutant, in panel C we show an alternative theory (in red) with a cutoff where only small steps are allowed. The actin monomers drawn below the top row are shaded according to the analytical theory results, with the darkest color normalized to the peak of the distribution.

The online version of this article includes the following video and figure supplement(s) for figure 4:

**Figure supplement 1.** Step distributions for myosin V and mutants with a freely rotating inter-leg joint.

**Figure 4—video 1.** BD simulation trajectory illustrating stepping for the 6IQ wild-type.

https://elifesciences.org/articles/51569#fig4video1

**Figure 4—video 2.** BD simulation of stepping for the 6IQ wild-type with a freely rotating joint, but with a longer step (final head separation > 36 nm).

https://elifesciences.org/articles/51569#fig4video2

**Figure 4—video 3.** BD simulation of stepping for the 6IQ wild-type with a freely rotating joint, but with a shorter step (final head separation < 36 nm).

https://elifesciences.org/articles/51569#fig4video3

**Figure 4—video 4.** Similar to *Figure 4—video 1*, but with an inter-leg joint constraint ($\mu_c = 5$).

https://elifesciences.org/articles/51569#fig4video4

**Figure 4—video 5.** Similar to *Figure 4—video 2*, but with an inter-leg joint constraint ($\mu_c = 5$).

https://elifesciences.org/articles/51569#fig4video5

**Figure 4—video 6.** Similar to *Figure 4—video 3*, but with an inter-leg joint constraint ($\mu_c = 5$).

https://elifesciences.org/articles/51569#fig4video6

for the 4IQ mutant. For 6IQ and 8IQ myosin V, the distributions are slightly broader, more similar to *Sakamoto et al. (2005)*. In particular, we capture the multi-peaked 8IQ distribution, with peaks at about 78 nm, 110 nm, and 135 nm (indicated by arrows in panel F). The first peak comes from steps in which the free leg detaches from the half-helical site behind the bound leg and steps to the half-helical site ahead of the bound leg. Similarly, the second peak corresponds to steps with one head-

to-head distance (during detachment or binding) being the full helical length and the other being the half-helical length. Finally, the third, smallest peak corresponds to the free leg starting and ending at the full helical distance. One might expect the rich stepping behavior of the 8IQ mutant to be inconsistent with the constrained diffusion hypothesis. If the joint does not freely rotate, can the myosin reach both the half- and full-helical actin subunits? Our model shows that a 5 $K_BT$ joint constraint energy is sufficiently weak to allow multi-peaked step distributions, but strong enough to considerably alter the diffusion (see previous section). We also tested large constraint energies and found that for $\mu_c k_B T \approx 10 - 12 k_B T$ we could no longer simultaneously fit the multi-peaked 8IQ step distribution and other step distributions. This upper bound on the joint constraint energy is similar to that estimated from diffusion contours.

The BD simulation step distributions are consistent with the results of the analytical model, as seen in *Figure 4*. Additionally, the simulations allow us to directly visualize examples of individual stepping trajectories. *Figure 4—video 1* through *Figure 4—video 3* show steps of three different sizes for the 6IQ wild-type without the inter-leg constraints ($\mu_c = 0$). *Figure 4—video 4* through *Figure 4—video 6* are analogous, but with the constraint present ($\mu_c = 5$).

## Myosin V exhibits robust step distributions under load

As a backward load force is applied to myosin V, forward steps decrease slightly in size while trailing leg stomps and backward steps become more likely. To elucidate this transition, we derive the combined leading and trailing leg step distribution (including stomps). To begin, in addition to the trailing leg steps considered above, we must also consider the distributions of steps and stomps originating from leading leg detachment. In this case, the bound leg position is behind the free leg starting position. So the initial distance between the heads is distributed as $\mathcal{P}^n_{-\text{dist}} = \mathcal{P}^{-n}_{\text{dist}}$. The leading leg step distribution is then given by the convolution $\mathcal{P}_{\text{L}}(z_n) = (\mathcal{P}_{-\text{dist}} * \mathcal{P}_{\text{L}})[n]$, with $z_n = n\Delta z/2$, and the combined leading/trailing leg step distribution is,

$$\mathcal{P}(z_n) = \frac{g}{1+g}\mathcal{P}_{\text{T}}(z_n) + \frac{1}{1+g}\mathcal{P}_{\text{L}}(z_n), \tag{6}$$

where $\mathcal{P}_{\text{T}}(z_n)$ is as defined in the preceding section. The force dependence of this distribution comes from the influence of the force on the myosin lever arm, which bends under the load giving rise to a new effective constraint direction $\hat{\mathbf{u}}'_c$ and new power stroke effectiveness $\mathcal{T}'$,

$$\hat{\mathbf{u}}'_c = \frac{\mathcal{T}\hat{\mathbf{u}}_c + \beta FL\hat{\mathbf{F}}}{\mathcal{T}'} \quad \text{and} \quad \mathcal{T}' = \sqrt{\mathcal{T}^2 + (\beta FL)^2 + 2\mathcal{T}\beta FL\hat{\mathbf{F}}\cdot\hat{\mathbf{u}}_c}, \tag{7}$$

where $F$ is the magnitude of the load force with direction $\hat{\mathbf{F}}$, $L$ is the myosin leg length, and $\beta = 1/(k_B T)$. For backward forces, $\hat{\mathbf{F}} = -\hat{\mathbf{z}}$, the lever arm bends toward the minus end of the actin increasing the effective constraint angle, but decreasing the power stroke effectiveness.

In *Figure 5A*, we show the combined distribution *Equation 6* for sub-stall backward forces, $F = 0.0$ and 1.0 pN, at stall force, $F = 1.9$ pN, and above stall, $F = 2.5$ pN. In this plot, peaks near +72 nm are forward steps, peaks near 0 nm are trailing or leading stomps, and peaks near −72 nm are backward steps. At zero force forward steps are dominant, exhibiting the experimentally observed step distribution, while forward leg stomps also occur less frequently. The myosin remains resilient as the force increases to 1 pN, with the step distribution shifting backward by approximately a single actin binding site. This result is consistent with the step distribution under 1 pN backward force measured by *Clemen et al. (2005)*, which was nearly identical to zero force step distributions.

Further increasing the force to stall at 1.9 pN, we see that the probability of forward steps dramatically decreases. Interestingly, even though forward steps are no longer kinetically dominant, the location and shape of their distribution remains robust, as shown in *Figure 5B* (BD simulations show the same qualitative behavior, see *Figure 5—figure supplement 1*). The stall force is defined to be the force at which the expected run length of the myosin is zero. Our calculation shows that stall is due to the emergence of a backward step distribution identical to the that for forward steps. Thus, at stall force the myosin takes forward and backward steps with equal probability, making no progress along the actin. In this regime, however, the dominant kinetic pathway is trailing leg stomps which give rise to the large peak around 0 in the step/stomp distribution. Finally, increasing the force

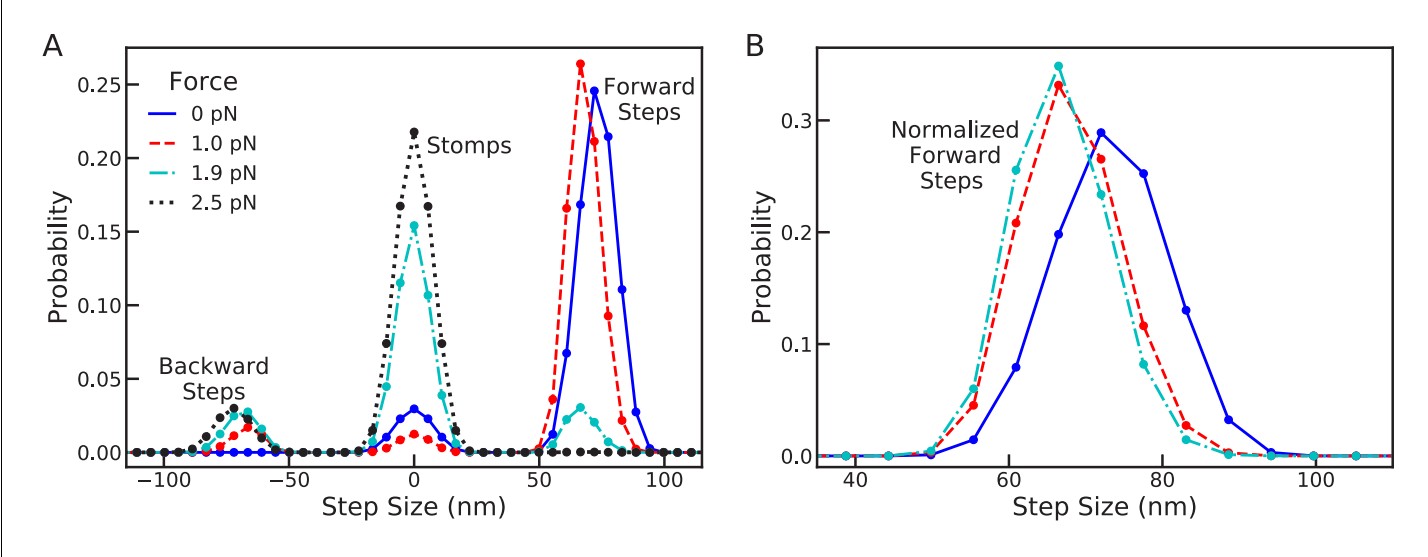

**Figure 5.** Changes in the full step distribution, including leading and trailing leg contributions, under backward load. (**A**) Distributions for zero force $F = 0$ pN (solid line), sub-stall force $F = 1$ pN (dashed line), stall force $F = 1.9$ pN (dot-dashed line), and super-stall force $F = 2.5$ pN (dotted line). The peaks near 72 nm, 0 nm and – 72 nm correspond to forward steps, stomps, and backward steps respectively. Applying force shifts the forward step distribution backward slightly (by about 1 actin subunit) and increases the probability of stomps and backward steps. (**B**) Normalized forward step distributions for $F = 0$ pN, $F = 1$ pN, and $F = 1.9$ pN. Even when other kinetic pathways are dominant the shape of the forward step distribution remains robust to load force.

The online version of this article includes the following figure supplement(s) for figure 5:

**Figure supplement 1.** Robust forward step distributions from Brownian dynamics simulations.

further eliminates all forward steps, increasing the prevalence of trailing stomps while shifting the backward step distribution by about one actin binding site.

An interesting prediction made by the model is the distribution of stomps, which is particularly apparent above stall force and has comparable width to the zero force step distribution. With ever improving imaging technology, it may soon be possible to reliably detect such variation in stomping experimentally. Because stomps are the dominant kinetic pathway in this force regime, many stomping events could be observed if sufficient imaging precision is achieved. Such measurements would provide further insights into how myosin V executes foot stomps. If the measured distribution significantly differs from our theoretical prediction, perhaps alternative mechanisms are involved in foot stomping. Since the leading head retains its ADP molecule after unbinding, it can in principle rebind quickly. Therefore, it is possible that leading foot stomps are dependent on non-equilibrium diffusion that occurs before the myosin legs relax to equilibrium. Further experiments are required to complete our understanding of myosin foot stomping.

The predictions made in *Figure 5* will require optical trap experiments in which a load force is applied to a cargo. If the contours of the distribution of the free head are to be simultaneously measured then it becomes necessary to also attach a gold nanoparticle (*Andrecka et al., 2015*). The analyses of the results of such experiments, which could test our predictions, might be complicated due to hydrodynamics effects of both the cargo and the gold nanoparticle.

## Probing the biological function of the joint constraint: effects on timing and consistency of stepping

In addition to making behavioral predictions, our model also provides insights into the potential biological function of the structural joint constraint. As discussed above, our computed diffusion contours and the measurements of *Andrecka et al. (2015)* indicate that the joint constraint reduces the diffusion search space from fully three-dimensional to nearly one-dimensional, with the free head locations concentrated along a curve. How does this effect influence the stepping behavior of

myosin V? One might expect that restricting the search space would affect both mean binding times and the step distributions.

Let us first consider the subject of binding times. As discussed above, binding in the model is more complex than just waiting for the head to diffuse within the capture radius $a$ of a binding site on actin. For the trailing leg, successful binding cannot occur until the head has hydrolyzed ATP and undergone the recovery stroke. Thus the mean timescale for ATP hydrolysis, $t_h$, sets the lower bound on the mean binding time. After the recovery stroke, if the trailing leg diffuses near one of the backward ($n<0$) binding sites, the probability of binding is reduced by a factor $b<1$ because the post-recovery-stroke head is not in an orientation that favors backward binding. Since polymer relaxation is fast between capture attempts, in effect this approximates partially absorptive reaction kinetics (*Šolc and Stockmayer, 1971*): the post-recovery-stroke head must make on average $1/b$ diffusive excursions near an $n<0$ site before it is captured. The net result of these two constraints, as derived in Appendix 1, is that the mean binding time for the trailing leg is:

$$t_T = r_T^{-1} + t_{\mathrm{h}}, \quad \text{where} \quad r_T = \sum_n b_n (t_{\mathrm{fp}}^n)^{-1}. \tag{8}$$

Here, $b_n = b$ for $n<0$, $b_n = 1$ for $n>0$. In contrast, if one is interested in the mean time for the head to diffuse within a distance $a$ of any of the binding sites, this is given by $t_{\mathrm{diff}} = \left[ \sum_n (t_{\mathrm{fp}}^n)^{-1} \right]^{-1}$ (irrespective of whether it is a trailing or leading head). *Figure 6* shows $t_T$ versus $t_{\mathrm{diff}}$ as a function of applied force F, and we always see $t_T > t_{\mathrm{diff}}$, as expected from the above constraints. For small $F$ the trailing leg kinetics is dominated by forward steps. $t_{\mathrm{diff}} < t_h$ in this regime, but even though the head can diffuse rapidly to forward binding sites, it does not bind until hydrolysis occurs. As the force is increased toward the stall regime, trailing leg stomps become the dominant pathway. The mean binding time $t_T$ increases by an order of magnitude, from about 1.6 ms at $F=0$ to a peak of 25 ms at $F=1.8$ pN. This is because the force biases the system toward backward binding sites (trailing leg stomps), but binding to those sites requires multiple diffusive attempts even after hydrolysis has occurred, because of the head orientation.

This increase may be reflected in an interesting experimental observation. *Veigel et al. (2002)* used an actin filament attached to beads in an optical trap, and studied the interactions between individual myosin V motors and the filament. By sinusoidally oscillating one bead (and hence the filament) they could use the amplitude of the resulting fluctuations to determine the stiffness of the motor-actin complex over time. Intervals of reduced stiffness were interpreted as times when only

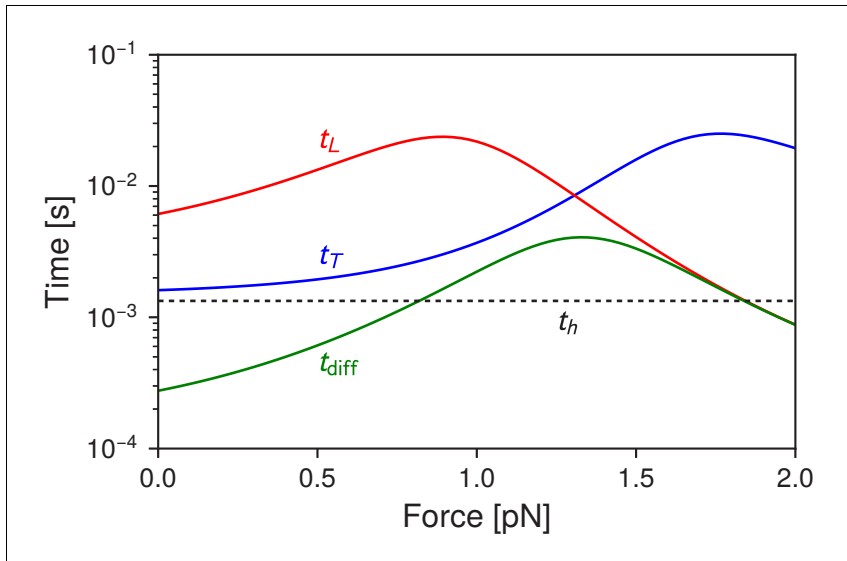

**Figure 6.** Myosin V timescales as a function of $F$, the backward load force. $t_{\mathrm{diff}}$ is the mean timescale for the detached head to diffuse within radius $a$ of any of the actin binding sites. $t_T$ and $t_L$ are the mean times for the trailing and leading heads to bind after detachment. For comparison, $t_h$ is the mean timescale of ATP hydrolysis.

one head was bound, and hence the unbound head is undergoing a diffusive search. At low loads ($F < 1$ pN) these intervals averaged around ~18 ms, while at near-stall loads the intervals lasted for hundreds of milliseconds. This order of magnitude relative increase is consistent with our prediction for $t_T$ as a function of $F$, with the interpretation that what they were observing at high loads was primarily trailing head stomping. The absolute timescales in the experiment are larger than in the theory (for example the predicted $t_T = 1.6$ ms at $F = 0$ versus the experimental value of ~18 ms), but this is likely due to the fact that the sinusoidal forcing (at 75 Hz with peak-to-peak amplitude 250 nm) is a substantial perturbation to the system that makes it harder for the head to bind (hence increasing mean binding times). However the relative increase of the binding times with force is captured by our model. Other experiments, where a gold nanoparticle is attached to one of the motor legs (*Dunn and Spudich, 2007*; *Andrecka et al., 2015*), have found binding timescales on the order of tens of milliseconds in the low load regime. Here there is another experimental artifact in play: as discussed in *Hinczewski et al. (2013)*, the drag from the gold nanoparticle can substantially increase diffusion times, with $t_T$ becoming much larger than $t_h$ at $F = 0$ because of slow diffusion.

To complete the description of the binding times, we note that the situation after leading leg detachment has several differences. Because the ADP molecule is retained, ATP hydrolysis is not necessary for rebinding, and the post power stroke orientation of the head favors backward rather than forward sites (with the latter penalized by a factor of $b$). The resulting leading leg mean binding time $t_L$ is:

$$t_L = r_L^{-1}, \quad \text{where} \quad r_L = \sum_n b_{-n}(t_{\text{fp}}^n)^{-1}. \tag{9}$$

At small loads the primary pathway for the leading leg is stomping. Since the head orientation necessitates multiple diffusive attempts to rebind for leading stomps, $t_L$ is much larger than $t_{\text{diff}}$ for small $F$, as seen in *Figure 6*. At larger loads backward stepping becomes the dominant pathway after leading leg detachment, and the combination of the backwards force and the favorable head orientation makes backward stepping nearly diffusion-limited, with $t_L$ approaching $t_{\text{diff}}$.

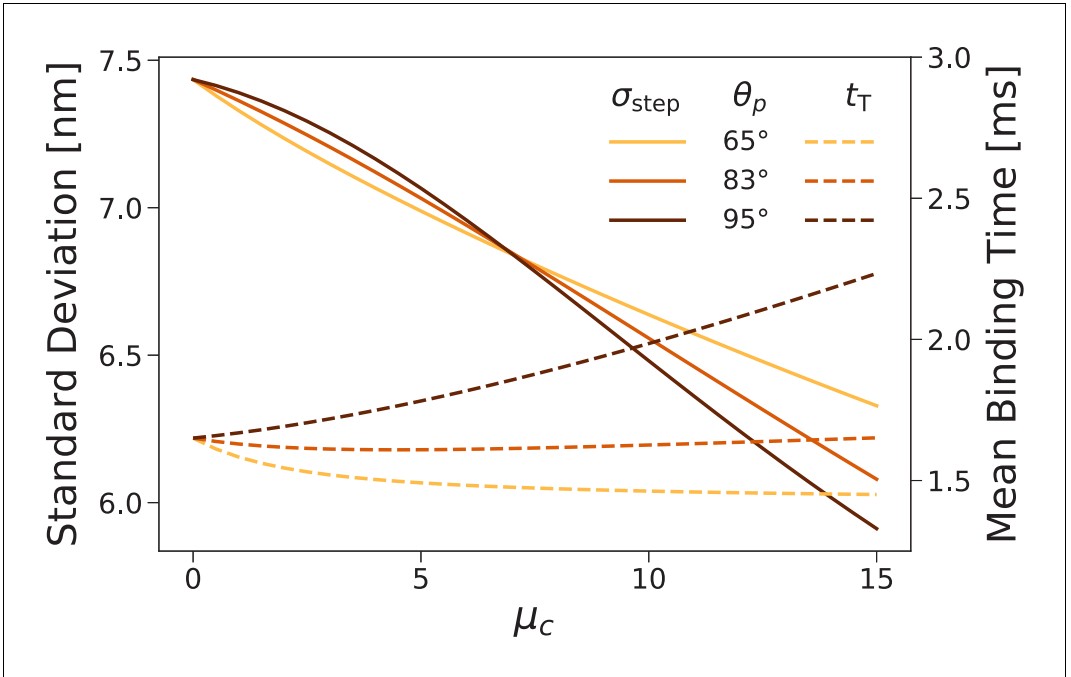

**Figure 7.** Forward step distribution width (solid lines) and mean binding time after trailing leg detachment (dashed lines) for $F = 0$ as a function of the inter-leg constraint strength $\mu_c$. We carried out this calculation for $\theta_p = 83°$ (the value used throughout this paper) as well as $\theta_p = 65°$ and $95°$. As the constraint is increased the step distribution narrows, while changes in the binding time are relatively small.

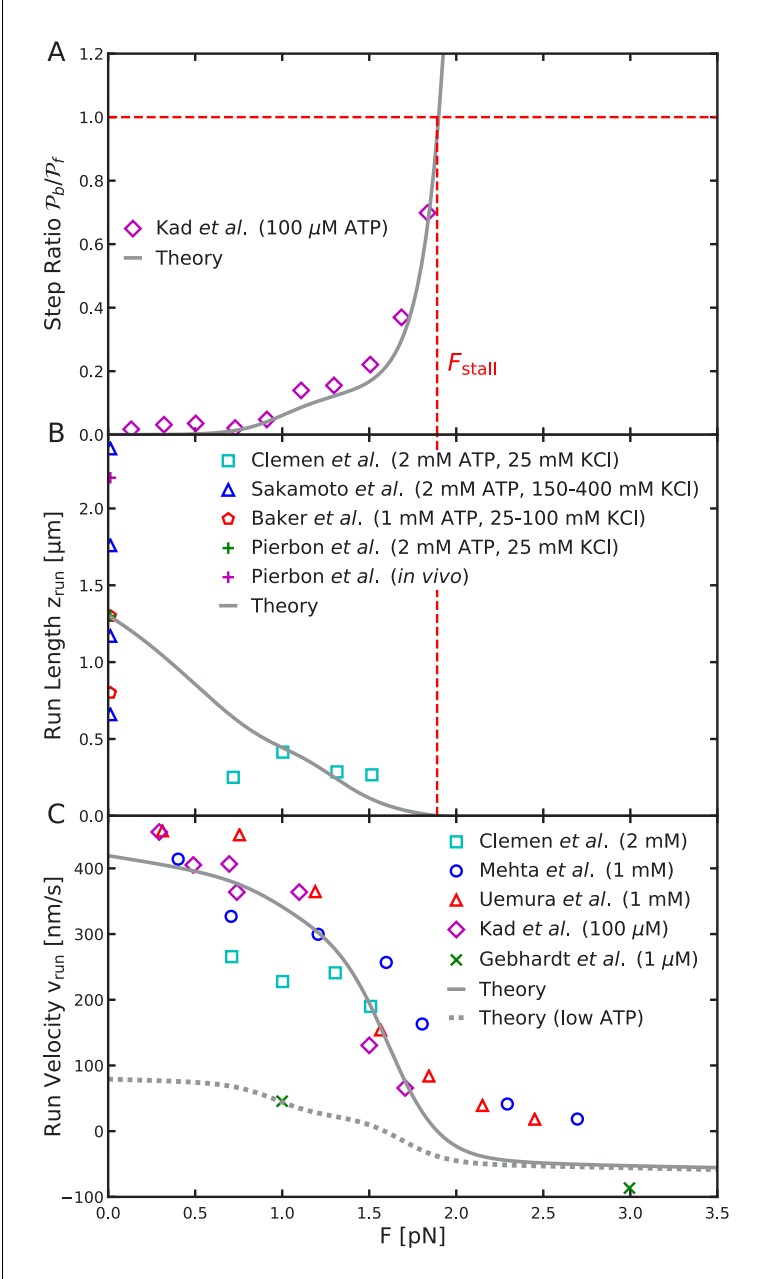

**Figure 8.** Load-dependent aspects of myosin V dynamics. (**A**) Backward-to-forward step ratio $\mathcal{P}_b/\mathcal{P}_f$; (**B**) mean run length $z_{\mathrm{run}}$; (**C**) mean run velocity $v_{\mathrm{run}}$. Analytical theory results are drawn as curves, experimental results as symbols. The legend symbols are the same as those in *Hinczewski et al. (2013)*, for ease of comparison, but the theory curves have been updated.

The online version of this article includes the following figure supplement(s) for figure 8:

**Figure supplement 1.** Load-dependence of step ratio, run length, and run velocity is captured by the free diffusion model.

Based on the above discussion of binding times, we can guess that the effect of the structural constraint $\mu_c$ might be relatively limited: while $\mu_c$ does indeed affect the diffusive search space, for forward stepping at low loads the binding timescale is dominated by the ATP hydrolysis time $t_h$. Despite the fact that increasing $\mu_c$ restricts the search space (and hence potentially makes diffusion faster), we find that $t_T$ at $F = 0$ does not substantially decrease and can increase depending on the preferred inter-leg angle (*Figure 7*, dashed lines). While adding the constraint makes it easier to find

a few target binding sites, it simultaneously makes it harder to find other sites, so that the overall change in binding time is small. Furthermore, we emphasize that head detachment (not the binding time) is the rate limiting action in the stepping cycle at low loads. Thus, the mean binding time only influences the motor dynamics indirectly, through determination of the mean number of steps taken before complete dissociation. To have any impact on motor performance, the mean binding time would have to decrease more substantially than we have found here.

So if the timing of the steps is not significantly affected, what about the step distributions? We computed these for several values of $\mu_c$ and found that increasing the joint constraint energy narrowed the distributions. *Figure 7* (solid lines) shows the standard deviation of the forward step distribution $\sigma_{\text{step}}$ as a function of $\mu_c$ for preferred inter-leg angle $\theta_p = 83°$ (the value used in above fits), 65°, and 95°. The standard deviation decreases, independent of angle preference, by approximately 0.1 nm per $k_B T$ of constraint energy. The constraint therefore plays a role in improving the consistency with which steps are made to a particular few target actin binding sites near the half helical length.

Finally, we conjecture that the joint constraint may also play a role in obstacle avoidance inside the crowded confines of a cell. Our diffusion contours (*Figure 3*, *Figure 3—figure supplement 1*, *Figure 3—figure supplement 2*) indicate that in the constrained diffusion model, the free head is significantly closer to the actin filaments throughout the diffusion. We believe that crowding could further restrict the conformational space explored by the free head, thus resulting in an even narrower step size distribution in vivo. Myosin V procession in the presence of obstacles would be an interesting topic for future study.

## Load dependence: step ratio, run length, and run velocity

For our final test of the constrained diffusion model we compare theoretical predictions to experiments on the load dependence of the forward-backward step ratio, the mean run length, and the mean run velocity (*Figure 8*). To fit the load dependence data, we require that the stall force and zero force run length agree with experimentally determined values, as further described below. This is achieved by varying the power stroke effectiveness $\mathcal{T}$, the binding penalty $b$, and capture radius $a$. All other parameters were held fixed as determined by experimental data or fitting the step distributions. As noted previously, we alternate between fitting the step distributions and force dependence to achieve optimal agreement between the theory and experiments.

### Step ratio

The backward-to-forward step ratio is defined as $\mathcal{P}_b/\mathcal{P}_f$, where $\mathcal{P}_b = 1/(1+g)\sum_{n<0} \mathcal{P}_L^n$ is the probability of taking a backward step and $\mathcal{P}_f = g/(1+g)\sum_{n>0} \mathcal{P}_T^n$ is the probability of taking a forward step. The step ratio measured by *Kad et al. (2008)* exponentially increases for large load force as the myosin transitions from forward to backward stepping. The step ratio $\mathcal{P}_b/\mathcal{P}_f = 1$ at $F = 1.9$ pN, when forward and backward steps are equally likely.

For the full range of physically reasonable parameters, we find that the force at which the step ratio is one is nearly identical to the stall force (defined as the force for which the run length $z_{\text{run}} = 0$). Therefore, as part of our fitting process, we require the stall force be $F_{\text{stall}} = 1.9$ pN. This value is within the range measured in other experiments. The agreement between the stall force and the force at which the step ratio is 1, is a consequence of the symmetry between the forward and backward step distributions shown in *Figure 5A*.

Interestingly, for intermediate forces beginning around $F \approx 1$ pN, there is a small increase in the step ratio before the exponential divergence near the stall force. We find that this effect is due to the binding penalty and only occurs for $b \lesssim 0.1$. Taking $b = 0.045$, the step ratio predicted by the model agrees well with the experimental data as shown in *Figure 8A*.

### Run length

The run length is the distance travelled by the myosin V along the actin in a given run. By averaging over the step distribution (*Equation 6*), we compute the mean run length,

$$z_{\text{run}} = \frac{1}{2} \frac{\sum_n z_n \mathcal{P}(z_n)}{\mathcal{P}_t(1 - \mathcal{P}_t)}, \tag{10}$$

where $\mathcal{P}_t$ is the termination probability and $z_n = n\Delta z/2$ as above. In this expression the numerator is the mean step size while the denominator accounts for the mean number of steps occurring in a run and normalization of the distribution $\mathcal{P}(z_n)$. In Appendix 1 we show that the mean number of steps is $\langle N_{\mathrm{run}} \rangle = 1/\mathcal{P}_t$. Finally, the factor of 1/2 accounts for the fact that the center of mass of the myosin moves half the distance of the head domain in a given step.

Experimental estimates, mostly performed with zero load force, report mean run lengths in the range 0.7 – 2.4 μm , with the large variation likely due to measurement conditions (*Sakamoto et al., 2000*; *Baker et al., 2004*; *Pierobon et al., 2009*). We choose a representative value $z_{\mathrm{run}} = 1.3$ μm for fitting the model. With this choice, the computed run length under force also is in reasonable agreement with measurements by *Clemen et al. (2005)* as shown in *Figure 8B*.

## Run velocity
The mean run velocity is $v_{\mathrm{run}} = z_{\mathrm{run}}/t_{\mathrm{run}}$, where $t_{\mathrm{run}}$ is the mean time elapsed during a processive run. In Appendix 1, we find

$$t_{\mathrm{run}} = \frac{\sum_n \mathcal{P}_T^n}{\mathcal{P}_t(1-\mathcal{P}_t)}\left(\frac{g}{1+g}t_{d1} + t_T\right) + \frac{\sum_n \mathcal{P}_L^n}{\mathcal{P}_t(1-\mathcal{P}_t)}\left(\frac{g}{1+g}t_{d1} + t_L\right), \tag{11}$$

where the mean trailing/leading binding times $t_T$ and $t_L$ are given by *Equations 8 and 9*. The other timescale in this expression $t_{\mathrm{wait}} = gt_{d1}/(1+g)$ is the mean waiting time between detachment events. Each time is weighted by the respective probability of occurrence and the mean number of steps taken.

The theoretical mean run velocity agrees well with the experimental data shown in *Figure 8C*. In addition to fitting the drop to zero run length at the stall force, our model also captures the gradual decrease in run velocity at low forces ($\leq 1\mathrm{pN}$), which is due to small backward shifts in the step distribution under load. This effect is not seen in the previous simplified model in which only half-helical steps were considered. We also compare to the large force measurements of *Gebhardt et al. (2006)*, which observed myosin V walking backward with velocity $\approx$ 90 nm/s under 3 pN backward force. This experiment was performed at 1 μM ATP concentration, which lowers the ATP binding rate and associated trailing leg dissociation rate. Therefore, for this comparison we set $t_{d1} = 2.2$ s$^{-1}$ as estimated from experimental kinetics (*Hinczewski et al., 2013*; *Gebhardt et al., 2006*). The low ATP run velocity is shown as the dashed curve in *Figure 8C*. The backward velocity under 3 pN force is comparable, though slightly smaller than the measured value. This small discrepancy, as well as disagreement with high force data (at 5 and 10 pN force), is likely due to additional kinetic pathways in the super-stall regime, such as the power stroke reversal discussed above.

## Robust procession under off axis loads
While we have focused on verifying the consistency of structurally constrained diffusion with the full range of experiments, our model also provides a fast analytical framework for studying complex behaviors of myosin V. As an example, we consider procession under off-axis load forces, which are not parallel to the axis of the actin filaments. Off-axis loads are likely especially relevant in vivo where the myosin navigates a crowded environment.

Experiments by *Oguchi et al. (2010)* measured the distance myosin V walked against off-axis loads applied by a bead in an optical trap. To apply the off-axis force, the actin was attached to a glass stage which was shifted perpendicularly to the actin axis after the myosin had taken two steps. For forces that were approximately ±25˚ off-axis on average, the myosin walked slightly further (~10 nm on average) than under backward force. The force at termination as well as the on-axis component of the termination force were also larger for off-axis loads. We emulated these experiments using Monte Carlo simulations, adjusting the load force after each step or stomp based on the distance from the starting location and shifting the force off axis after two steps. The average simulated run length and termination force each increased under off-axis loads (not shown), agreeing qualitatively with experiments.

We also directly compute the mean run length under a constant 1 pN load using *Equation 10* for a large range of off-axis forces parameterized by the spherical angles $\theta_F$ and $\phi_F$. The percent difference between the run length with off-axis force $(\theta_F, \phi_F)$ and the run length under backward force is shown in *Figure 9*. Our model estimates that at worst, changing the direction of a constant load

force can decrease the run length by ~15%, while many off-axis directions lead to considerable increases in run length. The shortest run length for 1 pN force occurs at ($\phi_F = 0$, $\theta_F \approx 20°$). This particular $\theta_F$ is slightly larger than that ($\theta_F \approx 6°$) which maximizes the effective constraint angle $\theta'_c = \arctan(\hat{\mathbf{x}} \cdot \hat{\mathbf{u}}'_c / \hat{\mathbf{z}} \cdot \hat{\mathbf{u}}'_c)$, but has a larger $\mathcal{T}'$, which strengthens the preference for locations along the constraint direction so that diffusion to forward actin sites becomes more difficult. The competition between these two effects leads to the minimized run length. For fully off-axis force ($\phi_F = \pm 90°$, $\theta_F > 0$), all run lengths are larger than those under backward force. These results further corroborate the robust processivity of myosin V under off-axis loads observed experimentally by *Oguchi et al. (2010)*.

## Discussion

### Constrained versus free diffusion: simplified effective theories

Our theoretical analysis presented above largely supports the constrained diffusion hypothesis. The $z - x$ diffusion contour produced by the constrained diffusion model closely resembles that measured by *Andrecka et al. (2015)* and we find strong quantitative agreement between the model predictions and experimental data for zero force step distributions and force dependence of the step ratio, run length, and run velocity. While this evidence is convincing, we emphasize that with relatively minor parameters changes the free diffusion model also makes similarly accurate predictions, with the exception of the diffusion contours. As is particularly apparent from comparison of the $z - x$ and $y - x$ projections (*Figure 3—figure supplement 2*), free diffusion doesn't produce multi-peaked distributions, implying that the freely rotating myosin explores the entire three dimensional space,

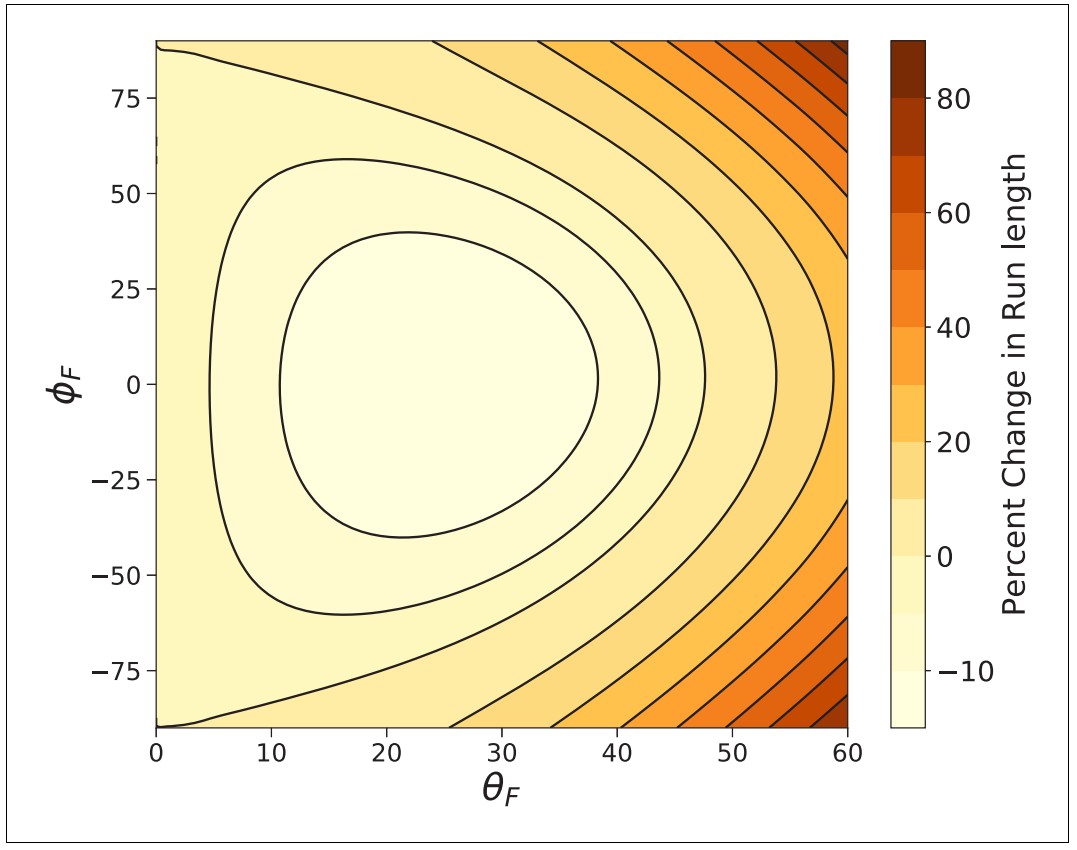

**Figure 9.** Myosin V run length under off-axis forces. Shown is the percent change in run length from that under backward force $z_{\text{run}}(\theta_F, \phi_F)/z_{\text{run}}(0,0) - 1$ computed using *Equation 9*. In the worst case ($\theta_F \approx 20°$, $\phi_F = 0$) the run length is decreased by ~15%. The run length most dramatically increases under fully off-axis forces ($\theta_F > 0$, $\phi_F = \pm 90°$).

contrary to the results of *Andrecka et al. (2015)*. With respect to other experiments, however, the free diffusion model performs favorably, agreeing with step distribution and force dependence data nearly as well as the constrained diffusion model (see *Figure 4—figure supplement 1* and *Figure 8—figure supplement 1*).

The similarity between the predictions of the constrained and free diffusion models can be explained through further analysis of the diffusion contours close to the actin. While the global diffusion patterns are very different from one another, close to the actin they look quite similar. Therefore, both the one-dimensional path under constrained diffusion and the full three-dimensional exploration under free diffusion favor binding to similar actin sites, namely those near the half-helical length of actin. This means that the free diffusion model accurately describes both step distributions and force dependence behavior. Since the diffusion time scale is much smaller than the head detachment time, the run velocity as a function of force is also quite similar between models.

The similarity between the free and constrained diffusion model predictions for on-actin behavior indicates that experimental measurements of such quantities (step distributions, run length, etc.) cannot effectively discern the true structure of the myosin V joint. The diffusive search, however, is dramatically influenced even by a small joint structural constraint. Therefore, to further probe the presence and consequences of structural constraints, experiments should directly measure the free leg diffusion. In particular, the $y - x$ projection of the diffusion shows particularly stark contrast between the constrained and free diffusion models. This contour corresponds to imaging the myosin from the front along the axis of the actin.

While our analysis and the most recent experiments suggest that myosin V has structurally constrained diffusion, the free diffusion model is still useful as a simplified effective theory for stepping behavior on the actin filaments. The precise values of physical parameters likely cannot be accurately determined with the free diffusion model, but extrapolation from fits to existing experimental data allows for valuable behavioral predictions. Both the free and constrained diffusion models can contribute to our understanding of the function of myosin V in complicated environments, for instance with off-axis load forces (see above). In fact, previous work (*Hinczewski et al., 2013*), indicates that the force dependence of average quantities, such as the forward step probability, step ratio, or run length, are well described by a further simplified model that only allows steps to the half-helical actin sites. Such a model cannot describe step distributions, but does contribute to our understanding of myosin V's resilience under backward force and robust motility under perturbations to structural parameters. Since our computed and simulated diffusion contours indicate a similarity between the constrained diffusion model and simplified free diffusion models near the actin, the latter remain quite useful for studying certain behaviors of myosin V.

## Conclusion

We have developed a comprehensive low-force model of myosin V, incorporating the polymeric nature of the lever arms, the joint angle preference which gives rise to a structurally constrained diffusive search, and the full set of kinetic pathways involving all actin binding sites. The analytical model allows us to compute bounds on the joint constraint energy and captures experimental results for step distributions and the force dependence of the step ratio, run length, and run velocity. While our results are largely in support of the constrained diffusion hypothesis, the theory also provides insight about how simplified models (eg. free diffusion) can provide a useful analytical description of some myosin V behaviors. Finally, using the model we can make predictions about experimentally measurable quantities including stomp distributions and robust run length under off-axis forces. To conclude we discuss limitations and potentially interesting extensions to the model for future studies.

Throughout the present study, we have assumed that myosin V walks along a single static actin filament. In reality, the many actin filaments within a cell can come together forming crossing and branching network structures (*Pollard et al., 2000*). Furthermore, bending and rotational fluctuations of the actin also occur and have been characterized experimentally (*Egelman et al., 1982*). A recent study observed myosin V walking on actin rafts (*Bao et al., 2013*), while fluctuations have been considered in previous theoretical work on step distributions (*Vilfan, 2005b*). New actin geometries and fluctuations could be easily incorporated in our model by convolving the distribution of actin binding site locations with the distribution for the position of the free myosin head.

One regime in which our model fails is under very large super-stall loads. In particular, the large backward run velocity measured by *Gebhardt et al. (2006)* at 5 pN and 10 pN backward force are not captured by the model. This alternative behavior is likely due to the power stroke reversal noted above and observed in experiments (*Sellers and Veigel, 2010*), which would promote more frequent large backward steps. It is also likely that under such extreme forces the detachment rates of the myosin heads are altered. By adding new power stroke reversal kinetic pathways, we expect our model to better capture the large force behavior of myosin V.

Finally, our methods may be more broadly applied to the large class of processive molecular motors that operate through the combination of chemical reactions and a diffusive Brownian search. Of particular interest are myosins VI and X, which operate through a similar set of kinetic pathways, but have heterogenous lever arms with both stiff and flexible components (*Sun and Goldman, 2011*). Perhaps applying our approach can resolve the controversy over the conformation of the myosin VI lever arm (*Spink et al., 2008*; *Mukherjea et al., 2009*; *Thirumalai and Zhang, 2010*). Beyond the myosin superfamily, we anticipate our approach may also prove useful for understanding kinesin and dynein motors as well as other biomolecules that can be suitably modeled by fluctuating polymer chains.

## Materials and methods

The theoretical methods used above are fully described in Appendix 1, while the details of the Brownian dynamics simulations are in Appendix 2.

## Acknowledgements

DH was supported by the Presidential Life Science Fellowship from Cornell and an NSF Graduate Research Fellowship (DGE-1650441). DT acknowledges support from the National Science Foundation (CHE 19–00033) and the Collie-Welch Chair (F-0019).

## Additional information

### Funding

| Funder | Grant reference number | Author |
| --- | --- | --- |
| National Science Foundation | DGE-1650441 | David Hathcock |
| National Science Foundation | CHE 19-00093 | Dave Thirumalai |

The funders had no role in study design, data collection and interpretation, or the decision to submit the work for publication.

### Author contributions

David Hathcock, Software, Formal analysis, Validation, Investigation, Visualization, Methodology; Riina Tehver, Conceptualization, Software, Formal analysis, Validation, Investigation, Visualization, Methodology; Michael Hinczewski, Conceptualization, Formal analysis, Supervision, Validation, Investigation, Visualization, Methodology, Project administration; D Thirumalai, Conceptualization, Supervision, Validation

### Author ORCIDs

David Hathcock ⓘD https://orcid.org/0000-0003-4551-9239
Riina Tehver ⓘD https://orcid.org/0000-0001-7406-3387
Michael Hinczewski ⓘD https://orcid.org/0000-0003-2837-7697

### Decision letter and Author response

Decision letter https://doi.org/10.7554/eLife.51569.sa1
Author response https://doi.org/10.7554/eLife.51569.sa2

## Additional files

### Supplementary files

• Source data 1. This ZIP file contains both the numerical data and Python scripts used to produce *Figure 3* through *Figure 9*, with individual directories corresponding to the materials for each figure.

• Transparent reporting form

### Data availability

All the data for the figures in the study (Figure 3–8), along with the corresponding code to process the data and produce the figures, is included in the source data file uploaded with the submission.

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

## Appendix 1

In this appendix we develop the analytical theory of myosin V. In the first two sections, we define the actin geometry and solve the first passage problem for binding to sites along the actin double helix. From this we derive step distributions, the backward-forward step ratio, the mean run length, and the mean run velocity. This analysis is quite general: it applies to any actin-based motor with a similar mechanochemical cycle, assuming one can compute the equilibrium probability density for the positions of the free head during diffusion. In later sections, we compute this probability density for myosin V, incorporating effects of an inter-leg constraint potential, volume exclusion, orientation requirements for binding, and load force. The combined result gives a complete diffusion-informed kinetic description of myosin V.

### Actin geometry

The actin double helix is composed of two filaments each containing 13 actin monomers per helical rotation. As it walks, the myosin heads can bind to any monomer along the actin strand. As discussed in the main text, the geometry of the actin helix is given by

$$\mathbf{r}_n = R(\cos\phi_n - 1)\hat{\mathbf{x}} + R\sin\phi_n\,\hat{\mathbf{y}} + (n/2)\Delta z\,\hat{\mathbf{z}}, \tag{12}$$

where $R$ = 5.5 nm is the radius of the helix, $\Delta z = 72/13 \approx 5.5\,\mathrm{nm}$ is the size of each actin monomer, and $\phi_n = -12\pi n/13$ is the angle between adjacent monomers (*Lan and Sun, 2006*). Even and odd $n$ respectively designate the positions of monomers on the 1st and 2nd filament of the double helix. In the following analysis, we assume the bound leg is attached to the $n = 0$ at position (0, 0, 0) on the first filament. Our work is a generalization of that by *Hinczewski et al. (2013)*, in which only the half helical binding sites $\mathbf{r}_{\pm 13}$ ($z = \pm 36\,\mathrm{nm}$) were considered.

### First passage time analysis

We begin by applying first passage time analysis to derive the stepping and stomping probabilities at each binding site along the actin. The first passage time of the free myosin leg to a given binding site along the actin $\mathbf{r}_n$ is approximately,

$$t_{\mathrm{fp}}^n \approx \frac{1}{4\pi a D_h \mathcal{P}(\mathbf{r}_n)}, \tag{13}$$

where $a = 0.4$ nm is the capture radius around the actin binding site, $D_h = 5.7 \times 10^{-7}$ cm²/s is the diffusion constant of the myosin V head, and $\mathcal{P}(\mathbf{r}_n)$ is the equilibrium probability distribution for the free myosin head evaluated at the actin binding site $\hat{\mathbf{r}}_n$. For a detailed calculation of this result using the renewal approach (*Van Kampen, 2007*), see *Hinczewski et al. (2013)*. The accuracy of this approximation relies on the assumption that $t_a = a^2/D_h \approx 0.18\,\mathrm{ns}$, the time scale describing the diffusion of the myosin head a distance $a$, is much less than the relaxation time of the myosin polymer legs $t_r \sim \mathcal{O}(1\,\mu\mathrm{s})$ which was calculated by *Hinczewski et al. (2013)* using Brownian molecular dynamics simulations (*Ermak and McCammon, 1978*).

The actin binding sites are located on the outer surface of each actin subunit. As discussed in the main text, we therefore require that the orientation of the free leg, defined in terms of its azimuthal orientation $\phi_f$, must be similar to that of the target actin binding site. Thus, for the description of binding events, we use the joint density $\mathcal{P}(\mathbf{r}_n, \delta\phi_1 > (\phi_f + \pi) - \phi_n > -\delta\phi_2)$. The two angles $(\delta\phi_1, \delta\phi_2)$ define a (possibly asymmetric) acceptance region, that is the deviations from the actin orientation $\phi_n = -12\pi n/13$ for which the head can still bind. The mean first passage time of the free head to a given binding site with the correct orientation is therefore,

$$t_{\mathrm{fp}}^n \approx \frac{1}{4\pi a D_h \mathcal{P}(\mathbf{r}_n, \delta\phi_1 > (\phi_f + \pi) - \phi_n > -\delta\phi_2)}. \tag{14}$$

This expression is used below in our derivations of physical observables for myosin V, connecting these quantities to mechanical properties of the polymer myosin lever arm.

While in the waiting state, both myosin heads are bound to the actin with ADP. The myosin can initiate a diffusive search through two kinetic pathways: (I) ADP is released from the trailing head (TH) and replaced by ATP, releasing the head from the actin, (II) the leading head (LH) detaches from the actin without ADP release. In the second scenario, dissociation of the LH occurs at a rate $t_{d2}^{-1}$ much less than $t_{d1}^{-1}$, the rate of ADP release/ATP binding in the first scenario. The probability of (I) and (II) occurring are then $g(1+g)^{-1}$ and $(1+g)^{-1}$ respectively, in terms of the gating parameter $g = t_{d2}/t_{d1} \gg 1$.

After the TH is released, it must hydrolyze ATP to execute the recovery stroke, which occurs at the rate $t_h^{-1}$. The free leg then diffusively searches for a binding site, reaching the capture radius of the $n^{\text{th}}$ site with rate $(t_{\text{fp}}^n)^{-1}$. Because the recovery stroke orients the myosin head to favor forward binding, we implement a binding penalty $b<1$ for sites behind the leading leg ($n<0$) designating the probability of binding once the site is reached. The binding penalty at site $n$ is denoted by,

$$b_n = \begin{cases} b, & n<0 \\ 1, & n>0. \end{cases} \tag{15}$$

With this definition, the effective diffusion/binding rate to each site is $b_n(t_{\text{fp}}^n)^{-1}$. Given the first passage times to each of the binding sites, we can calculate the distribution of free leg binding times to the $\mathbf{r}_n$ site,

$$
\begin{aligned}
f_T^n(t) &= \int_0^t dt'\, t_h^{-1} e^{-t'/t_h} b_n(t_{\text{fp}}^n)^{-1} e^{-(t-t')r_T} \\
&= b_n \frac{e^{-r_T t} - e^{-t/t_h}}{t_{\text{fp}}^n(1 - r_T t_h)},
\end{aligned} \tag{16}
$$

where $r_T = \sum_n b_n(t_{\text{fp}}^n)^{-1}$. Here the sum is over all binding sites which the myosin head can reach, that is those sites with $|\mathbf{r}_n|<2L$, where $L$ is the length of the myosin arms. The integral in *Equation 16* is the convolution of the probability of hydrolysis occurring by some time $t'$ and the probability of binding to the actin site over the interval time interval $t - t'$. These individual probabilities are modeled as exponential random variables with rate constants given by $(t_h)^{-1}$ and $b_n(t_{\text{fp}}^n)^{-1}$ respectively. The overall probability of binding to site $n$ given trailing head dissociation is:

$$\mathcal{P}_T^n = \int_0^\infty dt\, e^{-t/t_{d1}} f_T^n(t) = \frac{b_n t_{d1}^2}{t_{\text{fp}}^n(1 + r_T t_{d1})(t_{d1} + t_h)}, \tag{17}$$

where the exponential term implements the constraint that the free head must find a binding site before the bound head dissociates. From *Equation 16* we can also see that the average time to bind $t_T$ is independent of target site:

$$t_T = \frac{\int_0^\infty dt\, t f_T^n(t)}{\int_0^\infty dt\, f_T^n(t)} = r_T^{-1} + t_h. \tag{18}$$

The diffusive search of the LH is analogous with the following changes: (1) The LH retains its ADP molecule, so ATP hydrolysis is not required, (2) the LH is in the post power stroke orientation and favors binding to backward sites, behind the bound trailing leg. Thus, we obtain the LH binding probability and mean binding time by substituting $t_h \to 0$, $b_n \to b_{-n}$, $r_T \to r_L = \sum_n b_{-n}(t_{\text{fp}}^n)^{-1}$ in the above equations,

$$\mathcal{P}_L^n = \frac{b_{-n} t_{d1}}{t_{\text{fp}}^n(1 + r_L t_{d1})} \, t_L = r_L^{-1}.$$

From the two distributions, $\mathcal{P}_T^n$ and $\mathcal{P}_L^n$, we can calculate the probabilities of each kinetic pathway. The probability of forward steps, trailing stomps, leading stomps and backward steps are respectively

$$\mathcal{P}_f = \frac{g}{1+g} \sum_{n>0} \mathcal{P}_T^n, \quad \mathcal{P}_{Ts} = \frac{g}{1+g} \sum_{n<0} \mathcal{P}_T^n, \mathcal{P}_{Ls} = \frac{1}{1+g} \sum_{n>0} \mathcal{P}_L^n, \quad \mathcal{P}_b = \frac{1}{1+g} \sum_{n<0} \mathcal{P}_L^n,$$

where the factors $g(1+g)^{-1}$ and $(g+1)^{-1}$ account for the probability of trailing versus leading head detachment. The final pathway is termination, in which the free leg cannot complete its diffusive search before the bound leg detaches, leading to complete dissociation of the myosin from actin. The termination probability is simply, $\mathcal{P}_t = 1 - \mathcal{P}_f - \mathcal{P}_{Ts} - \mathcal{P}_{Ls} - \mathcal{P}_b$.

For comparisons to electron microscopy experiments (*Oke et al., 2010*), we require the distribution of head separation distances. The probability of observing bound myosin with $n$ actin subunits between the head is just the probability of taking a step/stomp of size $n$. This probability has contributions from each of the four primary kinetic pathways,

$$\mathcal{P}_{\text{dist}}^n = \frac{g}{1+g}(\mathcal{P}_T^n + \mathcal{P}_T^{-n}) + \frac{1}{1+g}(\mathcal{P}_L^n + \mathcal{P}_L^{-n}), \quad n>0. \tag{21}$$

The full step size distribution, measured by florescence imaging experiments (*Yildiz et al., 2003*; *Sakamoto et al., 2005*), is then the convolution of the binding distribution $\mathcal{P}_T^n$ with $\mathcal{P}_{\text{dist}}^n$. This gives,

$$\mathcal{P}_T(z_n) = \sum_{m>0} \mathcal{P}_{\text{dist}}^m \mathcal{P}_T^{n-m}, \tag{22}$$

where $z_n = n\Delta z/2$ and $\Delta z$ is the size of the actin monomers. Similarly, before the release of the LH, the distances are distributed according to *Equation 21*, but now the distance to the bound TH is negative, so substituting $\mathcal{P}_{\text{dist}}^m \rightarrow \mathcal{P}_{\text{dist}}^{-m}$ and summing over negative $m$, the full step size distribution for the leading head is given by,

$$\mathcal{P}_L(z_n) = \frac{1}{1+g}\sum_{m<0} \mathcal{P}_{\text{dist}}^{-m} \mathcal{P}_L^{n-m}. \tag{23}$$

For comparison to experiments, we only consider the trailing leg portion of step distribution, $\mathcal{P}_T(z_n)$, which we normalize appropriately. The combined leading and trailing leg step distribution, however, allows us to study the intricate changes in forward and backward steps/stomps in response to an applied load force. This behavior is highlighted in *Figure 5* in the main text. The required distribution is

$$\mathcal{P}(z_n) = \frac{g}{1+g}\mathcal{P}_T(z_n) + \frac{1}{1+g}\mathcal{P}_L(z_n), \tag{24}$$

where again trailing leg and leading leg events are weighted by the gating ratio. This distribution will also allow us to accurately calculate run length and velocity, accounting for the full distribution of step sizes and possibility of forward or backward motion due to stomping.

Given the termination probability $\mathcal{P}_t$ of each diffusive search, the average number of steps taken is then $\sum_{n=1}^{\infty} n(1-\mathcal{P}_t)^{n-1}\mathcal{P}_t = 1/\mathcal{P}_t$. Each step of size $n\Delta z/2$ occurs with probability $\mathcal{P}(z_n)/(1-\mathcal{P}_t)$. The total run length is defined as the change in position of the center of mass of the myosin, which is half the change in position of the free leg. Summing over all step sizes, we obtain

$$z_{\text{run}} = \frac{\Delta z \sum_n n\mathcal{P}(z_n)}{4\mathcal{P}_t(1-\mathcal{P}_t)}. \tag{25}$$

The average run velocity is defined as $v_{\text{run}} = z_{\text{run}}/t_{\text{run}}$, where $t_{\text{run}}$ is the average runtime. Before release of the TH or LH, the average waiting time bound to the actin is $t_{d1}t_{d2}/(t_{d1}+t_{d2}) = gt_{d1}/(1+g)$. After release, the mean binding times of the TH and LH are $t_T$ and $t_L$ respectively. The contributions of these times to the average run time is weighted by the sum of the TH/LH binding probabilities,

$$t_{\text{run}} = \frac{\sum_n \mathcal{P}_T^n}{\mathcal{P}_t(1-\mathcal{P}_t)}\left(\frac{g}{1+g}t_{d1}+t_T\right) + \frac{\sum_n \mathcal{P}_L^n}{\mathcal{P}_t(1-\mathcal{P}_t)}\left(\frac{g}{1+g}t_{d1}+t_L\right). \tag{26}$$

## Equilibrium probability of the free myosin head

The mean first passage time, $t_{\text{fp}}^n$ used in the preceding analysis relied on knowledge of the free leg equilibrium probability distribution, $\mathcal{P}(\mathbf{r}_n)$ evaluated at actin site $n$. In this section we derive the

general equilibrium probability $\mathcal{P}(\mathbf{r})$ for an arbitrary position $\mathbf{r}$. Projection of this distribution onto, for instance, the $x - z$ plane, gives a diffusion contour, like that measured by *Andrecka et al. (2015)*. Data from this experiment indicate that there may be an inter-leg joint potential $\mathcal{H}_J$ constraining the diffusion of myosin V. The precise form of the potential is discussed later, but we assume it is only a function of the inter-leg angle, or equivalently $\hat{\mathbf{r}}_f \cdot \hat{\mathbf{r}}_b$, where $\mathbf{r}_f$ and $\mathbf{r}_b$ are the end-to-end vectors for the free and bound legs. We also include a volume exclusion potential to capture the effects of steric repulsion between the myosin legs. Volume exclusion effects give rise to an effective repulsion between the bound and free heads given by the potential $\mathcal{H}_V = k_B T (d_V/r)^6$, where $r$ is the distance between the bound and free heads and $d_V$ is the effective length scale of the repulsion (see main text for more details). The probability can be obtained from the convolution of the end-to-end probability of the bound leg, $\mathcal{P}_b(\mathbf{r}_b)$ with that of the free leg $\mathcal{P}_f(\mathbf{r}_f)$, while accounting for the joint potential and volume exclusion through the appropriate Boltzmann factors,

$$\mathcal{P}(\mathbf{r}) = A\,e^{-(d_V/r)^6} \int d\mathbf{r}_b \int d\mathbf{r}_f\, \mathcal{P}_b(\mathbf{r}_b)\mathcal{P}_f(\mathbf{r}_f)\,e^{-\beta\mathcal{H}_J[\hat{\mathbf{r}}_f \cdot \hat{\mathbf{r}}_b]}\,\delta(\mathbf{r} - \mathbf{r}_b - \mathbf{r}_f), \tag{27}$$

where the constraint $\mathbf{r} = \mathbf{r}_b + \mathbf{r}_f$ is enforced with the $\delta$-function and $\beta = 1/(k_B T)$. Due to the volume exclusion term and because the joint constraint is a non-uniform weight in the convolution, we also require an overall normalization constant $A$. Each leg is modeled as a inextensible polymer with length L and persistence length $l_p$, and as noted above, the bound leg is assumed to be fixed at the origin $\mathbf{r} = 0$.

The free leg end-to-end equilibrium probability may be accurately approximated using polymer mean field theory (*Thirumalai and Ha, 1998*),

$$\mathcal{P}_f(\mathbf{r}_f) = A_f \xi_f^{-9/2} \exp\left(-\frac{3\kappa}{4\xi_f}\right), \tag{28}$$

where $\kappa = L/l_p$, $\xi_f = 1 - r_f^2/L^2$, and $A_f$ is a normalization constant. Because the polymer is inextensible the dimensionless parameter, $\xi_f$ is constrained to be between 0 and 1. The end-to-end vector $\mathbf{r}_f$ and be fully specified using $\xi_f$ as well as polar and azimuthal angles, $\theta_f$ and $\phi_f$. The normalization of the distribution then requires

$$\frac{L^3}{2} \int_0^1 d\xi_f (1 - \xi_f)^{1/2} \int_0^1 d\cos\theta_f \int_0^{2\pi} d\phi_f \mathcal{P}_f(\mathbf{r}_f) = 1, \tag{29}$$

which gives the normalization constant

$$A_f = \frac{9\sqrt{3}e^{3\kappa/4}\kappa^{7/2}}{8\pi^{3/2}L^3(3\kappa^3 + 12\kappa + 20)}. \tag{30}$$

For stiff polymers ($\kappa \approx 0$) relevant to myosin V, the distribution approximates a delta function at $r_f = L$. In the flexible regime ($\kappa \gg 1$), which may accurately describe motors with flexible components, the distribution becomes a spherically symmetric Gaussian centered at $\mathbf{r} = 0$.

The bound leg is assumed to be constrained in some direction $\hat{\mathbf{u}}_c$ by the power stroke. The tangent vector of this leg at the origin $\hat{\mathbf{u}}_0$ is therefore subject to a harmonic constraint with energy $\mathcal{H}_c = \frac{1}{2} k_B T \nu_c (\hat{\mathbf{u}}_0 - \hat{\mathbf{u}}_c)^2$, where there parameter $\nu_c$ is the strength of the constraint. This constraint biases the probability in the direction of $\hat{\mathbf{u}}_c$. We assume this effect is captured by the following ansatz:

$$\mathcal{P}_b(\mathbf{r}_b) = A_b \xi_b^{-9/2} \exp\left(-\frac{3\kappa}{4\xi_b} + \mathcal{T}\hat{\mathbf{u}}_c \cdot \hat{\mathbf{r}}_b\right), \tag{31}$$

where $\xi_b = 1 - r_b^2/L^2$, $A_b$ is a normalization constant, and $\mathcal{T}$ is a function of $\nu_c$ satisfying $\mathcal{T} = 0$ when $\nu_c = 0$ so that in the limit of no constraint, the distribution reduces to that of a free leg. With this addition, the normalization constant becomes

$$A_b = A_f \frac{\mathcal{T}}{\sinh\mathcal{T}}. \tag{32}$$

The function $\mathcal{T}$ was determined in *Hinczewski et al. (2013)* by matching the moments of this distribution to the exact moments for a semiflexible polymer under harmonic constraint. In the limit that $\kappa \ll 1$ and $\nu_c \gg 1$ (large stiffness and strong constraint), the result is

$$\mathcal{T} \approx 1 + \frac{20\nu_c}{20 + 7\kappa\nu_c}. \tag{33}$$

With the two probability distributions $\mathcal{P}_f(\mathbf{r}_f)$ and $\mathcal{P}_b(\mathbf{r}_b)$, we can now evaluate the convolution in *Equation 27*,

$$
\begin{aligned}
\mathcal{P}(\mathbf{r}) &= A\, e^{-(d_V/r)^6} A_f A_b \int d\mathbf{r}_b \int d\mathbf{r}_f \, \xi_f^{-9/2} \xi_b^{-9/2} \delta(\mathbf{r} - \mathbf{r}_b - \mathbf{r}_f) \\
&\quad \times \exp\left(-\frac{3\kappa}{4\xi_f} - \frac{3\kappa}{4\xi_b} + \mathcal{T}\hat{\mathbf{u}}_c \cdot \hat{\mathbf{r}}_b - \beta\mathcal{H}_J[\hat{\mathbf{r}}_f \cdot \hat{\mathbf{r}}_b]\right) \\
&= A\, e^{-(d_V/r)^6} A_f A_b \int d\mathbf{r}_f \, \xi_f^{-9/2} \xi_b^{-9/2} \exp\left(-\frac{3\kappa}{4\xi_f} - \frac{3\kappa}{4\xi_b} + \mathcal{T}\hat{\mathbf{u}}_c \cdot \hat{\mathbf{r}}_b - \beta\mathcal{H}_J[\hat{\mathbf{r}}_f \cdot \hat{\mathbf{r}}_b]\right).
\end{aligned} \tag{34}
$$

In the second line we integrate over the bound leg vector $\mathbf{r}_b$ with the delta function constraining $\mathbf{r}_b = \mathbf{r} - \mathbf{r}_f$ so that

$$\xi_b = 1 - \frac{r^2 + r_f^2 - 2rr_f\cos(\theta_f)}{L^2}, \qquad \hat{\mathbf{r}}_b = \frac{r\hat{\mathbf{r}} - r_f\hat{\mathbf{r}}_f}{\sqrt{r^2 + r_f^2 - 2rr_f\cos\theta_f}}, \tag{35}$$

where $\theta_f$ is the angle between the free leg end to end vector $\mathbf{r}_f$ and the target position $\mathbf{r}$.

The bound leg is constrained by the harmonic potential to the constraint direction, $\hat{\mathbf{u}}_c$, which has angular representation $(\theta_c, \phi_c = 0)$ with respect to the z-axis. To proceed, however, we consider the axis along the direction $\hat{\mathbf{r}}$. With respect to this axis, we can represent the unit vectors $\hat{\mathbf{r}}_f$ and $\hat{\mathbf{u}}_c$ by their polar and azimuthal angles $(\theta_f, \phi_f)$ and $(\tilde{\theta}_c, \tilde{\phi}_c)$. Then we may write the dot products as,

$$\hat{\mathbf{u}}_c \cdot \hat{\mathbf{r}}_b = \frac{r\cos\tilde{\theta}_c - r_f[\cos\theta_f\cos\tilde{\theta}_c + \cos(\phi_f - \tilde{\phi}_c)\sin\theta_f\sin\tilde{\theta}_c]}{\sqrt{r^2 + r_f^2 - 2rr_f\cos\theta_f}} \tag{36}$$

and

$$\hat{\mathbf{r}}_f \cdot \hat{\mathbf{r}}_b = \frac{r\cos\theta_f - r_f}{\sqrt{r^2 + r_f^2 - 2rr_f\cos\theta_f}}. \tag{37}$$

The integration element is then $d\mathbf{r}_f = r_f^2 dr_f d\cos\theta_f d\phi_f$ and after integrating over $\phi_f$, *Equation 34* becomes

$$
\begin{aligned}
\mathcal{P}(r\hat{\mathbf{r}}) &= 2\pi A A_f A_b \int_0^L r_f^2 dr_f \int_{-1}^1 d\cos\theta_f \, \xi_f^{-9/2} \xi_b^{-9/2} \exp\left(-\frac{3\kappa}{4\xi_f} - \frac{3\kappa}{4\xi_b} + \mathcal{T}_z \frac{r - r_f\cos\theta_f}{\sqrt{r^2 + r_f^2 - 2rr_f\cos\theta_f}}\right) \\
&\quad \cdot \exp\left(-\beta\mathcal{H}_J\left[\frac{r\cos\theta_f - r_f}{\sqrt{r^2 + r_f^2 - 2rr_f\cos\theta_f}}\right] - \left(\frac{d_V}{r}\right)^6\right) I_0\left(\frac{\mathcal{T}_x r_f\sin\theta_f}{\sqrt{r^2 + r_f^2 - 2rr_f\cos\theta_f}}\right),
\end{aligned} \tag{38}
$$

where $\mathcal{T}_z = \mathcal{T}\cos\tilde{\theta}_c$ and $\mathcal{T}_x = \mathcal{T}\sin\tilde{\theta}_c$. Performing the change of variables,

$$r_f \to L\sqrt{1 - \xi_f} \qquad \cos\theta_f \to \frac{r^2 + L^2(\xi_b - \xi_f)}{2rL\sqrt{1 - \xi_f}}, \tag{39}$$

we see that the main contributions to the integral in the stiff limit $\kappa \ll 1$ occur when $\xi_f \ll 1$ and $\xi_b \ll 1$. In this limit, the change of variables becomes $r_f \to L$, and $\cos\theta_f \to r/(2L)$ with Jacobian determinant $|\det J| = L^2/(4r)$ and upper integration limits $u = 2r/L - r^2/L^2$. Then *Equation 38* becomes

$$\mathcal{P}(r\hat{\mathbf{r}}) = \frac{L^4 \pi A A_f A_b}{2r} \int_0^u d\xi_f \int_0^u d\xi_b \, \xi_f^{-9/2} \xi_b^{-9/2} \exp\left(-\frac{3\kappa}{4\xi_f} - \frac{3\kappa}{4\xi_f} + \frac{\mathcal{T}_z r}{2L}\right)$$

$$\cdot \exp\left(-\beta\mathcal{H}_J[\frac{r^2}{2L^2}-1] - (d_V/r)^6\right) I_0(\mathcal{T}_x\sqrt{1-\frac{r^2}{4L^2}}). \tag{40}$$

After transforming back to the z-axis frame of reference, using $\cos\tilde{\theta}_c = \hat{\mathbf{r}} \cdot \hat{\mathbf{u}}_c$, the integral *Equation 40* evaluates to,

$$\mathcal{P}(r\hat{\mathbf{r}}) = \frac{8\pi L^4 A A_f A_b}{729\kappa^7 r}(20\sqrt{3\pi}u^{5/2}e^{\frac{3\kappa}{4u}}\mathrm{erfc}(\sqrt{\frac{3\kappa}{4u}}) + 3\sqrt{\kappa}(20u^2 + 10u\kappa + 3\kappa^3))^2$$

$$\cdot I_0(\mathcal{T}\sqrt{(1-(\hat{\mathbf{r}}\cdot\hat{\mathbf{u}}_c)^2)(1-\frac{r^2}{4L^2})})e^{\frac{1}{2}(\frac{\mathcal{T}\mathbf{r}\cdot\hat{\mathbf{u}}_c}{L}-\frac{3\kappa}{u})}e^{-\beta\mathcal{H}_J[\frac{r^2}{2L^2}-1]-(d_V/r)^6}. \tag{41}$$

This expression gives an analytical description of the free head diffusion as the myosin takes a step. Projecting the distribution onto a given plane produces a 2D diffusion contour, like that measured by *Andrecka et al. (2015)*. Depending on the joint constraint Hamiltonian and volume exclusion term, the normalization $A$ must generally be determined numerically. If $\mathcal{H}_J = 0$ and $d_V = 0$, then $A = 1$. In this case and for $n = \pm 13$, we have $\hat{\mathbf{r}}_{\pm 13} \cdot \hat{\mathbf{u}}_c = \pm\cos\theta_c$ and $u \approx 1$, so that the result of *Equation 41* reduces to that previously obtained for the half helical binding sites (*Hinczewski et al., 2013*).

## Joint constraint Hamiltonian

In light of experiments by *Andrecka et al. (2015)*, we use a joint potential $\mathcal{H}_J$ with an energy minimum at a certain preferred angle $\theta_p$. Instead of a freely rotating joint, this potential has an energy cost for deviations from $\theta_p$. Specifically we use the potential,

$$\mathcal{H}_J = \mu_c k_B T[1 - \cos(\theta_J - \theta_p)], \tag{42}$$

where $\mu_c$ is the dimensionless constraint strength and $\theta_J$ is the inter-leg angle at the joint. As discussed in the main text, this choice reduces to a harmonic potential when $\theta_J \approx \theta_p$. In large persistence length limit, $\kappa \ll 1$, we have $\cos\theta_J \approx -\hat{\mathbf{r}}_f \cdot \hat{\mathbf{r}}_b$. Above we found that in this limit, the potential becomes $\mathcal{H}_J[\hat{\mathbf{r}}_f \cdot \hat{\mathbf{r}}_b] \approx \mathcal{H}_J[r^2/(2L^2) - 1]$. Expanding the cosine in our potential we arrive at,

$$\mathcal{H}_J[r^2/(2L^2)-1] = \mu_c k_B T\left[1 - \left(1 - \frac{r^2}{2L^2}\right)\cos\theta_p - \frac{r}{L}\sqrt{1-\frac{r^2}{4L^2}}\sin\theta_p\right]. \tag{43}$$

## Orientation constraint for binding

The above model gives a description of the myosin diffusion: positions in the three dimensional space are visited by the free myosin head with probability $\mathcal{P}(r\hat{\mathbf{r}})$. To bind to actin, the free head must come sufficiently close to a binding size $\mathbf{r}_n$ and have the correct orientation, pointing toward the outer surface of actin subunit.

As discussed above, the density relevant for the first passage time to reach a binding site with the correct orientation is $\mathcal{P}_{\mathrm{bind}}(\mathbf{r}_n) = \mathcal{P}(\mathbf{r}_n, \delta\phi_1 > (\phi_f + \pi) - \phi_n > -\delta\phi_2)$. To evaluate this quantity we repeat the derivation from the previous section, but integrate $\phi_f$ only over the acceptance region $(\phi_n - \delta\phi_2, \phi_n + \delta\phi_1)$. Note we can choose our axes so that $\tilde{\phi}_c = 0$. In principle we should use $\tilde{\phi}_n$, the azimuthal orientation of actin subunit $n$ with respect to the $\hat{\mathbf{r}}_n$ axis. The sites with non-negligible probability of binding, however, have $|z| \gg |x|, |y|$, so that $\tilde{\phi}_n \approx \phi_n$. Carrying out the derivation, we arrive at

$$\mathcal{P}_{\mathrm{bind}}(\mathbf{r}_n) = \frac{8\pi L^4 A A_f A_b}{729\kappa^7 r_n}(20\sqrt{3\pi}c^{5/2}e^{\frac{3\kappa}{4c}}\mathrm{erfc}(\sqrt{\frac{3\kappa}{4c}}) + 3\sqrt{\kappa}(20c^2 + 10c\kappa + 3\kappa^3))^2 e^{\frac{1}{2}(\frac{\mathcal{T}\mathbf{r}_n\cdot\hat{\mathbf{u}}_c}{L}-\frac{3\kappa}{c})}$$

$$\cdot e^{-\beta\mathcal{H}_J[\frac{r^2}{2L^2}-1]-(d_V/r)^6} \int_{\phi_n-\delta\phi_2}^{\phi_n+\delta\phi_1} d\phi_f \exp\left(\cos\phi_f \mathcal{T}\sqrt{(1-(\hat{\mathbf{r}}_n\cdot\hat{\mathbf{u}}_c)^2)(1-\frac{r^2}{4L^2})}\right). \tag{44}$$

To evaluate this probability the constant $A$ must be calculated numerically and the integral over $\phi_f$ is computed numerically for each target binding site. This quantity gives us the first passage times in *Equation 14* and allows for computation of step distributions and other physical observables derived above.

## Equilibrium probability of the free myosin head under load

If we apply a load force $\mathbf{F}$ exerted at the end of the bound the myosin leg, the distribution *Equation 31* is multiplied by a factor $\exp(\beta F r_b \hat{\mathbf{F}} \cdot \hat{\mathbf{r}}_b)$. This term is simply the Boltzmann factor describing the energy cost required for the myosin leg to move a distance $r_b$ in the direction $\hat{\mathbf{r}}_b$ under the load force. Again assuming the stiff limit, $(r_b \approx L)$, this expression becomes $\exp(\beta F L \hat{\mathbf{F}} \cdot \hat{\mathbf{r}}_b)$. Thus, the load force simply changes the effective constraint direction and strength, $\hat{\mathbf{u}}_c \rightarrow \hat{\mathbf{u}}'_c$ and $\mathcal{T} \rightarrow \mathcal{T}'$ where

$$\hat{\mathbf{u}}'_c = \frac{\mathcal{T}\hat{\mathbf{u}}_c + \beta F L \hat{\mathbf{F}}}{\mathcal{T}'} \quad \mathcal{T}' = \sqrt{\mathcal{T}^2 + (\beta F L)^2 + 2\mathcal{T}\beta F L \hat{\mathbf{F}} \cdot \hat{\mathbf{u}}_c}. \tag{45}$$

Making these substitutions in *Equation 44*, gives the equilibrium free-leg probability of being at a binding site with the orientation required to bind for myosin V under load force $\mathbf{F}$.

## Appendix 2

### Brownian dynamics simulations

In order to explore the dynamics of a myosin motor and explicitly incorporate excluded volume and other constraints, we used Brownian Dynamics (BD) simulations. In the simulations, myosin is modeled as a chain of beads of diameter $d_M$ = 2 nm. The wild-type myosin V dimer has two lever arms that include 6 IQ domains each. We model the dimer as a polymer of N = 35 beads. To study how the lever arm length affects the step size distributions, we created myosin V models with 4 IQ domains and 8 IQ domains. The models consisted of 25 and 45 beads, respectively.

The Hamiltonian that governs the interactions between the constituent beads of myosin V and myosin-actin interactions in the BD simulations is given as:

$$H = H_{bond} + H_{repel} + H_{bend} + H_{AM,repel} + H_{AM,bind} \tag{46}$$

The five contributions to the Hamiltonian are listed below. First, the bond potential between myosin V beads is given as:

$$H_{bond} = \frac{1}{2}k_s \sum_{i=1}^{N-1}(r_{i,i+1} - d_M)^2, \tag{47}$$

where $r_{i,i+1}$ is the center-to-center distance between the beads $i$ and $i+1$. The bond strength in our simulations is $k_s$ = 700 kcal/ mol/ nm². In practice, $k_s$ must be large enough to prevent the contour length of the polymer from changing significantly. Next,

$$H_{repel} = \epsilon \sum_{i=1}^{N-2}\sum_{j=i+2}^{N}\left(\frac{d_M}{r_{i,j}}\right)^6 \tag{48}$$

where $r_{i,j}$ is the distance between two beads. This term ensures that there are no unphysical overlaps between any beads. In the equation above, $d_M$ = 2 nm which corresponds to the diameter of a bead. The potential strength $\epsilon$ = 1 kcal/mol.

A similar term, $H_{AM,repel}$, is included to ensure that a myosin lever arm is sterically repulsed by the actin filament. For $H_{AM,repel}$, the potential has the a similar form as *Equation 48* with the parameter, $d_{AM}$ = 5.5 nm (the sum of the radii of a bead and the actin filament) and the analog to $r_{i,j}$ is calculated as the perpendicular distance from the center of a lever arm bead to the actin filament.

Actin-myosin binding is given by a screened Coulomb potential,

$$H_{AM,bind} = -\frac{V_0}{\rho}e^{-\alpha\rho} \tag{49}$$

where the potential strength, $V_0$ = 3.3 kcal/mol, the screening length $1/\alpha$ = 0.5 nm, and $\rho$ is the distance between the binding region on the stepping head of myosin V and any actin binding site. The actin binding region of the myosin motor is located on the surface of the last bead. We consider myosin bound and a trajectory complete when the binding energy becomes larger than 2.0 kcal/ mol. We ran 200 stepping trajectories for each different setup we studied. Each trajectory ended when the stepping head was bound at an actin binding site.

The bending potential is given by:

$$H_{bend} = \sum_{i=1}^{N-1}k_{bend,i}(d_M^2 - \vec{r}_{i,i+1}\cdot\vec{r}_{i-1,i}) \tag{50}$$

where the bending constant, $k_{bend,i}$ is bond specific. Incorporating a heterogeneous bending constant allows us to create a flexible joint in the middle of the polymer and introduce varying amount of flexibility throughout the polymer structure. The bending constant is related to the persistence

length $L_p$ of a polymer as: $k_{bend} = L_p k_B T / d_M^3$, where $k_B T$ determines the thermal energy scale. The values of $k_{bend}$ used in the simulations are given in the table below.

**Bead sequence number**

| 6 IQ model | 4 IQ model | 8 IQ model | $k_{bend}$ in kcal/mol |
|---|---|---|---|
| 1 | 1 | 1 | 150 |
| 2 | 2 | 2 | 34.9 |
| 3–17 | 3–12 | 3–22 | 23.7 |
| 18 | 13 | 23 | 0 |
| 19–33 | 14–23 | 24–43 | 23.7 |
| 34 | 24 | 44 | 34.9 |
| 35 | 25 | 45 | N/A |

## Power stroke

The first term in the bending potential, $k_{bend,1}(d_M^2 - \vec{r}_{1,2} \cdot \vec{r}_{0,1})$, is used to create a power stroke by including a vector, $\vec{r}_{1,0}$ that makes an angle $\theta_c = 65°$ with the actin filament. Specifically, the first term in the bending potential is zero when the angle between the vector created by the first two beads and the imposed direction is zero; the energy penalty increases as the attached lever arm direction deviates from the direction imposed by constraint angle, $\theta_c$. In practice, a large $k_{bend,1}$ keeps the bound head in the post-powerstroke orientation.

## External force

To analyze the effect of external pulling force, we add a constant force in the $-\hat{z}$ direction to the myosin lever arm junction (i. e. the middle bead). The other forces and parameters remain unchanged.

## Joint constraint

To study a model for myosin V that steps in a compass-like motion, we introduce an additional harmonic constraint between the two beads adjacent to the junction. We report data where the spring constant of the constraint is 5 $k_B T / nm^2$ and the equilibrium length of the spring corresponds to a 90° angle between the two lever arms.

## Glass cover

To investigate the effect of a cover slip (or any other excluded space) on the dynamics of myosin V dimer, we excluded half of the search space using a soft-core repulsive potential similar to *Equation 48*): $H_{glass} = \epsilon \sum_{i=1}^{N} \left( \frac{d_C}{r_i} \right)^6$. The data reported in this paper was obtained using $d_C$ = 1 nm, $\epsilon$ = 5 kcal/mol, and $r_i$ is measured as a perpendicular distance from a bead to the glass plate that is directly under the actin filament in the $xz$-plane.

