## [Decision Letter]

**Acceptance summary:**

This manuscript revisits the modeling of the stepping mechanics of the processive actin-dependent motor, myosin V, by introducing a structural constraint wherein the angle between the lever arms connecting the two heads is relatively fixed as suggested by a more recent experimental paper (Andrecka et al., 2015). That paper suggested that, rather than function as a freely jointed swivel the junction between the two lever arms was relatively fixed and that the free head of myosin V did not explore random space while searching for a new, forward actin binding site. Rather, it occupied an off-actin dwell space from which it periodically explored the actin landscape. The new model introduces this structural constraint and examines the consequences in terms of free head localization, selection of forward actin monomer binding site, step size and step directionality. In addition, the authors examine the effect of modeled load on several parameters, including the effect of off axis load. The new model predicts the experimental free head localization data obtained in Andrecka et al., 2015, as well as the kinetic and mechanical data found in many other myosin V papers.

**Decision letter after peer review:**

Thank you for submitting your article "Myosin V executes steps of variable length via structurally constrained diffusion" for consideration by *eLife*. Your article has been reviewed by two peer reviewers, and the evaluation has been overseen by a Reviewing Editor and Aleksandra Walczak as the Senior Editor. The following individuals involved in review of your submission have agreed to reveal their identity: Anatoly Kolomeisky (Reviewer #1); James R. Sellers (Reviewer #2).

The reviewers have discussed the reviews with one another and the Reviewing Editor has drafted this decision to help you prepare a revised submission.

Summary:

This manuscript revisits the modeling of the stepping mechanics of the processive actin-dependent motor, myosin V, by introducing a structural constraint wherein the angle between the lever arms connecting the two heads is relatively fixed as suggested by a more recent experimental paper (Andrecka et al., 2015). That paper suggested that, rather than function as a freely jointed swivel the junction between the two lever arms was relatively fixed and that the free head of myosin V did not explore random space while searching for a new, forward actin binding site. Rather, it occupied an off-actin dwell space from which it periodically explored the actin landscape. The new model introduces this structural constraint and examines the consequences in terms of free head localization, selection of forward actin monomer binding site, step size and step directionality. In addition, they examine the effect of modeled load on several parameters, including the effect of off axis load. The conclusion is that their new model predicts the experimental free head localization data obtained in Andrecka et al., 2015, as well as the kinetic and mechanical data found in many other myosin V papers. However, they add the comment that a freely diffusional model also fits most of the data in the literature with the exception of the discrete location for the dwell state.

This is a comprehensive theoretical work on uncovering mechanism of the motion of myosin-V motor proteins. The subject is very important, and this work makes a huge step in our understanding of this difficult problem. The paper is very nicely written and all the details are clearly explained. The authors are noted for their self-critical approach and for critically analyzing many experimental and theoretical results. For example, the discussion of asymmetry in diffusion contours is very honest and very objective. They combine computer simulations and analytical theory, the latter of which providing a better molecular understanding. This work should stimulate a lot of new experiments.

Essential revisions:

1) The use of mean first-passage times as estimates for the binding reaction processes assumes that the reaction is taking place after a single collision (diffusion limit). While this might be the case, it is not proven in the case of motor proteins. I would add a caution at this point about the application of mean first-passage times. By the way, this might also be the reason why the theory does not work at large forces. Some discussion might be useful here. For example, authors should note that Dunn et al., 2007, Andrecka et al., 2015, and Veigel et al., 2002, all estimated the actual transit time from detachment of the trail head to its reattachment at a new forward position to be much greater than the theoretic first passage time.

2) Figure 5. The Veigel et al., 2002 paper showed evidence of stiffness changes at stall forces that may represent the proposal of trail head stomping. While the duration of these events were not quantified in this paper, lifetimes of 200—500 ms are shown in Figure 5 of that paper at 100 μm ATP. Could the authors comment on whether their model would predict such long lifetimes of single-headed attachment?

---

## [Author Response]

Essential revisions:1) The use of mean first-passage times as estimates for the binding reaction processes assumes that the reaction is taking place after a single collision (diffusion limit). While this might be the case, it is not proven in the case of motor proteins. I would add a caution at this point about the application of mean first-passage times. By the way, this might also be the reason why the theory does not work at large forces. Some discussion might be useful here. For example, authors should note that Dunn et al., 2007, Andrecka et al., 2015, and Veigel et al., 2002, all estimated the actual transit time from detachment of the trail head to its reattachment at a new forward position to be much greater than the theoretic first passage time.

The reviewers raise an important point that was not sufficiently clear in the original manuscript. Binding in our model is not simply diffusion-limited, but this was not discussed explicitly in the main text, and we did not show any binding time results. For that reason we have added a new section to the main text, “Probing the biological function of the joint constraint: effects on timing and consistency of stepping”. Part of this section is devoted to a detailed description of binding timescales, along with a new figure (Figure 6) that shows a comparison of diffusion, hydrolysis, and binding timescales as a function of load force. Quoting from that new section:

“As discussed above, binding in the model is more complex than just waiting for the head to diffuse within the capture radius *a* of a binding site on actin. […] Figure 6 shows *t_T_
*versus *t*_diff_ as a function of applied force *F*, and we always see *t_T_ > t*_diff_, as expected from the above constraints.”

The discussion of the experimental estimates for transit times is later in the same section, and we quote it in response to the next reviewer comment.

2) Figure 5. The Veigel et al., 2002 paper showed evidence of stiffness changes at stall forces that may represent the proposal of trail head stomping. While the duration of these events were not quantified in this paper, lifetimes of 200—500 ms are shown in Figure 5 of that paper at 100 μm ATP. Could the authors comment on whether their model would predict such long lifetimes of single-headed attachment?

This is indeed an interesting experimental observation which we believe is linked to trailing leg stomping. We now discuss this experiment in our new section on timing that we have added to the main text. Quoting from that section:

“For small *F* the trailing leg kinetics is dominated by forward steps. *t*_diff_
*< t_h_
*in this regime, but even though the head can diffuse rapidly to forward binding sites, it does not bind until hydrolysis occurs. […] Here there is another experimental artifact in play: as discussed in Hinczewski et al.,2013, the drag from the gold nanoparticle can substantially increase diffusion times, with *t_T_
*becoming much larger than *t_h_
*at *F* = 0 because of slow diffusion.”